# Towards Synthesizing Complex Programs from Input-Output Examples

**Xinyun Chen   Chang Liu   Dawn Song**
University of California, Berkeley

## Abstract

In recent years, deep learning techniques have been developed to improve the performance of program synthesis from input-output examples. Albeit its significant progress, the programs that can be synthesized by state-of-the-art approaches are still simple in terms of their complexity. In this work, we move a significant step forward along this direction by proposing a new class of challenging tasks in the domain of program synthesis from input-output examples: learning a context-free parser from pairs of input programs and their parse trees. We show that this class of tasks are much more challenging than previously studied tasks, and the test accuracy of existing approaches is almost 0%.

We tackle the challenges by developing three novel techniques inspired by three novel observations, which reveal the key ingredients of using deep learning to synthesize a complex program. First, the use of a non-differentiable machine is the key to effectively restrict the search space. Thus our proposed approach learns a *neural program* operating a domain-specific *non-differentiable machine*. Second, recursion is the key to achieve generalizability. Thus, we bake-in the notion of recursion in the design of our non-differentiable machine. Third, reinforcement learning is the key to learn how to operate the non-differentiable machine, but it is also hard to train the model effectively with existing reinforcement learning algorithms from a cold boot. We develop a novel two-phase reinforcement learning-based search algorithm to overcome this issue. In our evaluation, we show that using our novel approach, neural parsing programs can be learned to achieve 100% test accuracy on test inputs that are $500\times$ longer than the training samples.

## 1 Introduction

Learning a domain-specific program from input-output examples is an important open challenge with many applications (Balog et al., 2017; Reed & De Freitas, 2016; Cai et al., 2017; Li et al., 2017; Devlin et al., 2017; Parisotto et al., 2017; Gulwani et al., 2012; Gulwani, 2011). Approaches in this domain largely fall into two categories. One line of work learns a neural network (i.e., a fully-differentiable program) to generate outputs from inputs directly (Vinyals et al., 2015b; Aharoni & Goldberg, 2017; Dong & Lapata, 2016; Devlin et al., 2017). Despite their promising performance, these approaches typically cannot generalize well to previously unseen inputs. Another line of work synthesizes a non-differentiable (discrete) program in a domain-specific language (DSL) using either a neural network (Devlin et al., 2017; Parisotto et al., 2017) or SMT solvers (Ellis et al., 2016). However, the complexity of programs that can be synthesized using existing approaches is limited.

Although many efforts are devoted into the field of neural program synthesis, all of them are still focusing on synthesizing simple textbook-level programs, such as array copying, Quicksort, and a combination of no more than 10 string operations. We believe that the next important step for the community is to consider more complex programs.

In this work, we endeavor to pursue this direction and move a big step forward to synthesize more complex programs than before. Along the way, we identify several novel challenges dealing with complex programs that have not been fully discussed before, and propose novel principled approaches to tackle them. First, an end-to-end differentiable neural network is hard to generalize, and in some cases is hard to even achieve a test accuracy that is greater than 0%. We observe that a neural network is too flexible to approximate any functions, but the programs that we want to synthesize typically lie

in a search space of interest. It is very hard to restrict the learned neural network to always represent an instance in the search space. When not, the network is simply overfitting to the training data, and thus cannot generalize. To mitigate this issue, we employ the approach to train a differentiable *neural program* to operate a domain-specific *non-differentiable machine*. This combination allows us to restrict the search space by defining the non-differentiable machine, so that any neural program that can operate the machine is always a valid program of interest.

Second, the domain-specific machine needs to be expressive enough. In particular, state-of-the-art approaches, such as RobustFill (Devlin et al., 2017), may fail at tasks involving long outputs because they can only synthesize programs of up to 10 string operations, which do not support *recursion*. As also noted by Cai et al. (2017), recursion is a key concept to enable perfect generalization. Therefore, it is desirable that the non-differentiable machine can bake-in the concept of recursion into the design.

Third, the non-differentiable machine makes the model hard to be trained end-to-end, especially when the traces to operate the machine are not given during the training. Thus, we rely on a reinforcement learning algorithm to train the neural program while recovering the execution traces. However, this is challenging. Previous attempts (Zaremba & Sutskever, 2015) along this direction can only succeed to learn to compute addition of two numbers, and fail even for the tasks of three-number additions. In our evaluation, we observe that training from a cold start is the main difficulty. In particular, the model trained using existing reinforcement learning algorithms from a cold boot always gets stuck at a local minimum to fit to only a subset of the training samples. More importantly, the recovered traces are sometimes "wrong", which aggravates the issue. To the best of our knowledge, this issue is a long-standing challenging problem for neural program synthesis, and we are not aware of any promising solution. To tackle this issue, we propose a *two-phase reinforcement learning-based algorithm*. Intuitively, we break up the whole problem into two separate tasks: (1) searching for possible traces; and (2) training the model with the supervision of traces. Each of these two tasks are easier for reinforcement learning to handle, and we develop a nested algorithm to combine the solutions to these two tasks to provide the final solution.

To demonstrate our ideas, we propose a novel challenging problem: learning a program to parse an input satisfying an (unknown) context-free grammar into its parse tree. As we will show, this class of programs are much more challenging to learn than those considered previously: using most state-of-the-art approaches, the test accuracy almost remains 0% when the test inputs are longer than the training ones. Meanwhile, learning a parser is also an important problem on its own with many applications, such as easing the development of domain-specific languages and migrating legacy code into novel programming languages. In this sense, this problem exhibits both more complexity and more practicality than some previously considered problems, such as synthesizing an array-copying function. Therefore, our newly proposed problem serves as a good next step challenge to tackle in the domain of learning programs from input-output examples.

To implement the idea of learning a neural program operating a non-differentiable machine, we first design the *LL machine*, which bakes in the concept of recursion in its design. We also design a *neural parsing program*, such that every neural parsing program operating an LL machine is restricted to represent an LL parser. To evaluate our approach, we develop two programming languages, an imperative one and a functional one, as two diverse tasks. Combined with our newly proposed two-phase reinforcement learning-based algorithm, we demonstrate that for both tasks, our approach can achieve 100% accuracy on test samples that are $500\times$ longer than training samples, while existing approaches' corresponding test accuracies are 0%.

To summarize, our work makes the following contributions: (1) we propose a novel strategy that combines training neural programs operating a non-differentiable machine with reinforcement learning, and this strategy allows us to synthesize more complex programs from input-output examples than those that can be handled by existing techniques; (2) we reveal three important observations why synthesizing complex programs can be challenging; (3) we propose a novel two-phase reinforcement learning-based algorithm to solve the cold start training problem, which may be of independent interests; (4) we propose the parser learning problem as an important and challenging next step for program synthesis from input-output examples that existing approaches fail with 0% accuracy; (5) we demonstrate that our strategy can be applied to solve the parser learning problem with 100% accuracy on test samples that are $500\times$ longer. We consider applying our strategy to more complex tasks other than the parser learning problem as future work.

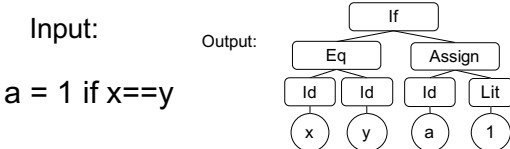

Figure 1: An input-output example. The input is a sequence of tokens [“a", “=", “1", “if", “x", “==", “y"], and the output is its parse tree. The non-terminals are denoted by boxes, and terminals are denoted by circles.

## 2 THE PARSING PROBLEM AND APPROACH OVERVIEW

To illustrate our strategy towards synthesizing complex programs, we want to put our presentation in a context. To this end, in this section, we define the parsing problem and outline our approach. Note that our strategy could also be adapted for other problems. In the following, we start with the formal definition of the parsing problem.

**Definition 1 (The parsing problem)** *Assume there exist a context-free language $\mathcal{L}$ and a parsing oracle $\pi$ that can parse every instance in $\mathcal{L}$ into an abstract syntax tree. Both $\mathcal{L}$ and $\pi$ are unknown. The problem is to learn a parsing program $P$, such that $\forall I \in \mathcal{L}, P(I) = \pi(I)$.*

Figure 1 provides an example of an input and its output parse tree. The internal nodes of the tree are called *non-terminals*, and the leaf nodes are called *terminals*. The sets of non-terminals and terminals are disjoint. Each terminal must come from one input token, but the non-terminals do not necessarily have such a correspondence. To simplify the problem, we assume the input is already tokenized. The sets of all non-terminals and terminals can be extracted from the training corpus, i.e., all nodes in the output parse trees of training samples. In this work, we assume the vocabulary set (i.e., all terminals and non-terminals) is finite, and our work can be extended to handle unbounded vocabulary set with techniques such as pointer networks (Vinyals et al., 2015a).

**Remarks.** Note that our parsing problem has its counterpart to handle natural languages, which has been extensively studied in the literature (Andor et al., 2016; Chen & Manning, 2014; Yogatama et al., 2016; Dyer et al., 2016). We want to remark on the difference between the two problems. On the one hand, a programming language with a context-free grammar is easier to learn than a natural language in the sense that the grammar has a rigorous specification to avoid ambiguity. That is, it is always possible to construct a parser to achieve 100% accuracy for a context-free programming language, while this may not be the case for a natural language. On the other hand, learning a programming language parser may be more challenging than a natural language parser, since an instance in a programming language can be arbitrarily long, while a natural language sentence typically has only a limited number of words. In this sense, an approach that can learn a natural language parser well may not be able to handle a programming language, when the test samples are much longer than training samples. As we will observe in our evaluation, this is indeed an issue for existing approaches.

We also want to remark on potential applications of the parsing problem. Nowadays, we need to develop new domain-specific languages in many scenarios. While the current practice is to develop the grammar and parser manually, this process is error-prone. In our experience, when creating the datasets for evaluation, we find that designing a training set of (program, parse tree) pairs is typically easier, but developing the parser takes $2\times$ or $3\times$ more time than developing the training set. Intuitively, building a training set only needs developing a tutorial including basic examples whose parse trees are easy to construct manually; on the other hand, implementing a parser requires much longer time in debugging, typically with the help of the developed training set. Therefore, our proposed parser generation problem is a novel practical use case of program-by-example.

**Challenges.** Learning the parsing program $P$ is challenging for several reasons. First, the correspondence between non-terminals and input tokens is unknown. For example, in Figure 1, the parser needs to find out that token “=" corresponds to the non-terminal Assign. Second, the order of non-terminals in the tree may not align well with the input tokens. For example, in Figure 1, the sub-expression “a=1", which is to the left of the sub-expression “x==y", corresponds to the

right child of the non-terminal `If`, which is to the right of the sub-tree corresponding to "`x==y`" (the sub-tree whose root is the non-terminal `Eq`). Third, the association of tokens may depend on other tokens. For example, in expressions "`x+y*z`" and "`x+y+z`", whether "`x+y`" forms a sub-tree depends on the operator (i.e., "+" or "*") after it.

**Solution overview and paper organization.** To tackle the challenges, we make several innovations. First, we employ a paradigm to learn a differentiable *neural program* operating a *non-differentiable* machine in Section 3. In particular, we introduce *LL machines* (Section 3.1) as an example of non-differentiable machine. LL machines can restrict that a learned program always represents a program of interest, i.e., an LL parser.

To bake-in the notion of recursion, we think the non-differentiable machine should have a stack structure and provide `CALL` and `RETURN` instructions to simulate recursive calls. In addition, the neural program should operate the machine based on only the stack top to make sure the learned program can generalize. We design the LL machine and *neural parsing programs* (Section 3.2) to operate an LL machine following these principles. We will show that in doing so, the neural parsing program can be learned to generalize to longer programs. Note that here we mainly use the LL machines and the neural parsing program to demonstrate that a design supporting recursion is necessary to achieve generalization, and our strategy is not limited to this combination only.

Third, we design a novel two-phase reinforcement learning-based search algorithm (Section 4) to tackle the challenge of training a neural parsing program without the supervision of execution traces. In our evaluation (Section 6), we demonstrate that our approach achieves 100% training accuracy, while the accuracies on all testsets are also 100%. We then discuss related work in Section 7, and conclude in Section 8.

## 3 Neural Programs operating a Non-differentiable Machine

In this section, we demonstrate how to employ the neural program operating a non-differentiable machine approach to tackle the neural parsing problem. To carry out this agenda, we first design *LL machines* in Section 3.1. The design bakes in the notion of recursion. That is, an LL machine has a stack for recursive call stacks, and provides a `CALL` instruction and a `RETURN` instruction which can be used to simulate recursive calls.

Then, we design a neural program operating the LL machine in Section 3.2. We show that the neural program makes its decisions based on only the stack top in the LL machine. In doing so, the neural program can take advantage of the recursion support of an LL machine, and be learned to generalize to handle longer inputs. We present the details in the following.

### 3.1 LL Machines

We briefly present the design of LL machines. It is inspired by the $LL(1)$ parsing algorithm (Parr & Quong, 1995), and an LL machine is generic enough to allow construct any LL parsers. For example, in our evaluation, we demonstrate that both an imperative language and a functional language can use the LL machine to construct their parsers.

The LL machine maintains an internal state using a stack of *frames* and an external state of the stream of input tokens. Each stack frame is an ID-list pair, where the ID is a *function ID* and in the list are $(n, T)$ pairs, where $T$ is a parse tree, and $n$ is the root node of $T$.

An LL machine has five types of instructions: `SHIFT`, `RETURN`, `CALL`, `REDUCE`, and `FINAL`. Their semantics are presented in Table 1. Intuitively, these instructions can be classified into three classes. The first class, including `SHIFT` and `REDUCE`, takes charge of all parser related primitives. In fact, Knuth (1965) has demonstrated that `SHIFT` and `REDUCE` primitives are sufficient to construct any context-free grammars.

The second class, including `CALL` and `RETURN`, is used to operate the stack to simulate recursive calls. As we will show in the next section, the controller, i.e., a neural parsing program, can decide what instruction to execute based on only the stack top frame, whose size can be bounded. This is crucial for the well-trained neural parsing program to generalize to test samples that are much longer than training samples.

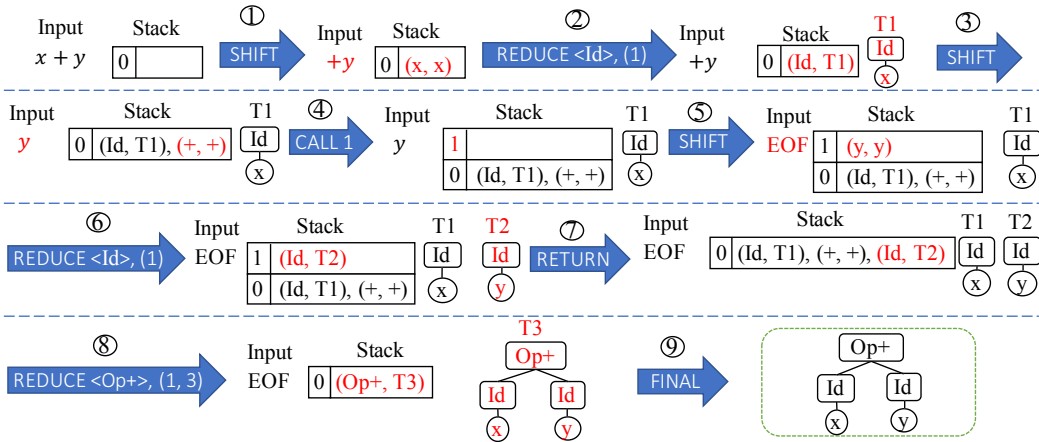

Figure 2: A full example to parse `x+y` into a parse tree. 9 instructions are executed to construct the final tree. We use red labels to illustrate the changes after performing each operation. For ease of illustration, if a tree has only one node, which is a terminal, then we simply use this terminal to represent both the root node and the tree in the stack; otherwise, we draw the tree next to the stack, and refer to it with a unique label in the stack.

| Insturction | Argument | Description |
|---|---|---|
| SHIFT | (None) | Pull one token from the input stream to append to the end of the stack top. |
| REDUCE | $n, c_1, ..., c_m$ | Reduce the stack's top frame to form a new node rooted at $n$. $c_i$ denotes that the $i$-th child of the root $n$ is the $c_i$-th element originally in the stack top frame. |
| CALL | $fid$ | Push one frame with $(fid, [])$ at the stack top. |
| RETURN | (None) | Pop the stack top and append the popped data to the new stack top. |
| FINAL | (None) | Terminate the execution, and return the stack top as the output. |

Table 1: LL machine instruction semantics

The third class, including only FINAL, is an instruction to terminate the machine's execution and produce the final result. This instruction should be included in any non-differentiable machine design.

Through this design, we want to highlight three key properties in the design of a non-differentiable machine: instructions for the core functionality related to the programs of interest; instructions to enable recursion; and instructions to produce the results and terminate the machine. We present a running example using LL machine in Figure 2, and more details and examples to explain how an LL machine works can be found in Appendix B.

## 3.2 A Neural Parsing Program

A parsing program operates an LL machine via a sequence of LL machine instructions to parse an input to a parse tree. Specifically, a parsing program operating an LL machine decides the next instruction to execute after each timestep. A key property of the combination of the LL machine and the parsing program is that its decision can be made based on three components only: (1) the function ID of the top frame; (2) all root nodes of the trees (but not the entire trees) in the list of the top frame; and (3) the next input token. We can safely assume that the list in any stack frame can have at most $K$ elements (see Appendix B). Here, $K$ is a hyper-parameter of the model. Therefore, the parser only needs to learn a (small) finite number of scenarios in order to generalize to all valid inputs.

To learn the parsing program, we represent it as a neural network, which predicts the next instruction to be executed by the LL machine. Specifically, we consider two inference problems that compute the probabilities of the type and arguments of the next instruction respectively:

$$p(inst|fid, l, tok) \qquad\qquad p(\arg_{inst}|inst, fid, l, tok)$$

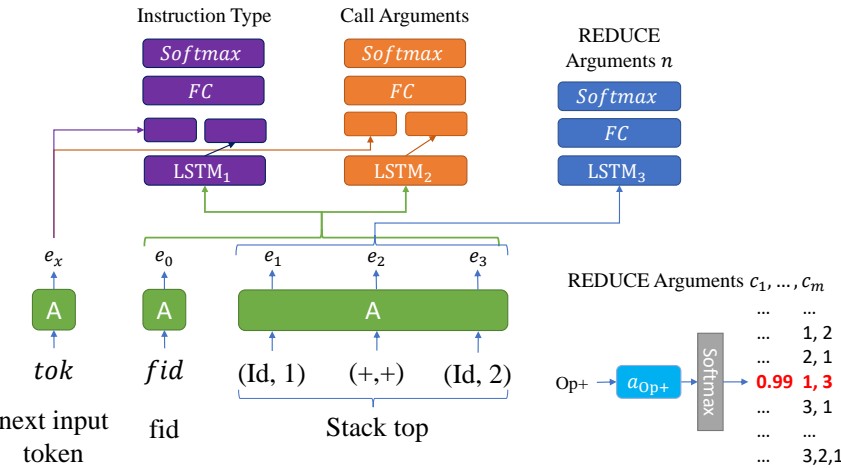

Figure 3: The Neural Parsing Program Model.

where $(\mathit{fid}, l)$ denotes the current stack top, $\mathit{tok}$ is the first token of current input, $\mathit{inst}$ is the type of the next instruction, $\arg_{\mathit{inst}}$ are the arguments of the next instruction $\mathit{inst}$. Note that the second probability is needed only if the predicted instruction type is either CALL or REDUCE.

At a high-level, at each step, the neural network first converts each root node in the list of the top stack frame into an embedding vector, and then runs three separate LSTMs to predict the type (i.e., Formula (1)) and arguments (i.e., Formula (2) and (3)) of the next instruction. This network is illustrated in Figure 3. We explain each component in the following. For notation, we use $L$ to denote the length of $l$, and $D$ the dimensionality for both input embeddings and LSTM hidden states. $\mathbf{softmax}(...)_i$ denotes the $i$-th dimension of the softmax output. More subtle details can be found in Appendix C.

**Embeddings.** For each element $(n_i, T_i)$ $(1 \leq i \leq L)$ in the stack top's list $l$, we use a lookup table $A$ over all terminals and non-terminals to convert $n_i$ into the embedding space. Specifically, we compute a $D$-dimensional vector $e_i = A(n_i)$ for $1 \leq i \leq L$. Thus, we compute $e_1, ..., e_L$ from $l$.

**Instruction probability.** We use an LSTM to compute $P_{\mathbf{inst}}(\mathit{inst}|\mathit{fid}, l, \mathit{tok})$ as follows:

$$P_{\mathbf{inst}}(\mathit{inst}|\mathit{fid}, l, \mathit{tok}) = \mathbf{softmax}(W_1 \cdot \mathrm{LSTM}_1(A(\mathit{fid}), e_1, ..., e_L) + W_2 \cdot A(\mathit{tok}))_{\mathit{inst}} \quad (1)$$

Specifically, each function ID $\mathit{fid}$ is treated as a special token, which is converted into the embedding using $A$ as well. We use $\mathrm{LSTM}_1(A(\mathit{fid}), e_1, ..., e_L)$ to indicate the final hidden state of $\mathrm{LSTM}_1$ when the input sequence to the LSTM is $A(\mathit{fid}), e_1, ..., e_L$. Further, $A(\mathit{tok})$ encodes the current token using the same lookup table $A$ as above. $W_1$ and $W_2$ are $M \times D$ trainable matrices, where $M = 5$ since there are 5 different types of instructions. The operation $W_1 \cdot \mathrm{LSTM}_1(A(\mathit{fid}), e_1, ..., e_L) + W_2 \cdot A(\mathit{tok})$ is equivalent to concatenating the LSTM output with $A(\mathit{tok})$ and passing it through a fully-connected layer (i.e., $FC$ in Figure 3).

**Predicting CALL arguments.** To predict argument $\mathit{fid}'$ of the CALL instruction, we compute

$$P_{\mathbf{fid}}(\mathit{fid}'|\mathit{fid}, l, \mathit{tok}) = \mathbf{softmax}(W_1' \cdot \mathrm{LSTM}_2(A(\mathit{fid}), e_1, ..., e_L) + W_2' \cdot A(\mathit{tok}))_{\mathit{fid}'} \quad (2)$$

This part is similar to the one for next-instruction prediction as shown in Formula (1), though a different set of parameters (i.e., $W_1', W_2'$) is used. The lookup table $A$ is the only overlap.

**Predicting REDUCE arguments.** For a REDUCE instruction, we need to predict both $n$ and $(c_1, ..., c_m)$, which define how to construct the new sub-tree. To achieve this, the model predicts $n$ first, and then predicts $(c_1, ..., c_m)$ based on $n$. Specifically, we have

$$P_{\mathbf{n}}(n|l) = \mathbf{softmax}(W'' \cdot \mathrm{LSTM}_3(e_1, ..., e_L))_n \quad (3)$$

$$P_{\mathbf{c}}(c_1, ..., c_m|n) = \mathbf{softmax}(a_n)_{c_1, ..., c_m} \quad (4)$$

where $\text{LSTM}_3$ is the third LSTM, $W''$ is an $N \times D$ trainable matrix. Here $N$ is the number of different types of non-terminals. Note that different from predicting the next instruction type and the `CALL` arguments, predicting $n$ does not look at $fid$ and $tok$, only $e_1, ..., e_L$.

The choice of $c_1, ..., c_m$ is entirely decided by $n$. To this end, we convert this prediction problem as a one-hot prediction problem. In particular, we encode each possible combination of $c_1, ..., c_m$ into a unique ID. In fact, given $m \leq K$, there are at most $f(K) = \lfloor K! exp(1) - 1 \rfloor$ possible different combinations of $c_1, ..., c_m$, where $K!$ is the factorial of $K$, and $exp(1)$ is the base of the Natural Logarithm (see Appendix C). Therefore, we model the prediction problem of $c_1, ..., c_m$ as a $f(K)$-way classification problem.

In Equation 4, $a_n$ is a $f(K)$-dimensional trainable vector for each $n$. Assume the ID for $(c_1, ..., c_m)$ is $\xi$, then $\textbf{softmax}(...)_{c_1,...,c_m}$ indicates the $\xi$-th dimension of the softmax output. Notice that setting $K$ to 4 is enough to handle two non-trivial languages used in our evaluation. In both cases, $f(K) \leq 65$, which is tractable as the number of classes in a classification problem. We consider to handle a larger $K$ as future work.

## 4 LEARNING A NEURAL PARSING PROGRAM

Training a neural parsing program is challenging due to the non-differentiability of the LL machine. The main problem is that the execution trace of the LL machine is unknown, and thus reinforcement learning is necessary for recovering the execution trace. However, training with reinforcement learning is very unstable, and is usually stuck at a local minimum that can fit to only a few examples. In such a case, more importantly, the recovered execution traces may be "wrong". To the best of our knowledge, this is a long-standing open problem, and there is no effective mechanism to find the global optimum for a model that can fit to all examples at once.

In this work, we tackle this challenge by proposing a two-phase training strategy. In fact, the main challenge of training with a reinforcement learning algorithm lies in the difficulty of jointly recovering the execution traces and learning a model with effective parameters. The main idea of our training strategy is to decouple the problem into two phases, where the first phase tries to recover the execution traces, while the second phase tries to train a set of parameters. In the following, we first explain the high-level idea of this two-phase training approach (Section 4.1), and then explain how reinforcement learning is used (Section 4.2). We will illustrate why and how our algorithm works with a running example in Section 5.

### 4.1 TWO-PHASE TRAINING STRATEGY

When the training set contains only input-output pairs without any information on the execution traces, learning a model that can parse all valid inputs 100% accurately is challenging. The main issue is that a learned model may correctly parse some inputs, but fail on others. We observe that for each input-output pair, there may exist multiple valid execution traces (see Appendix D for an example), where a model trained to mimic one certain trace for one input-output pair may not be able to learn to mimic one certain execution trace for another pair at the same time. Thus, our goal is to find *consistent* execution traces for all input-output pairs in the training set.

To achieve this goal, we learn the neural parsing program in two phases. First, for each input-output pair, we find a set of valid candidate instruction type traces with a preference toward shorter ones. We refer to this set of traces as the *candidate trace set* for a given input-output pair. Second, we try to search for a *satisfiable specification*. A *specification* is a set of input-output-trace triples that assign an instruction type trace from the corresponding candidate trace set for each input-output pair in the training set. We say that a specification is *satisfiable*, if there exists a neural parsing program that can parse all inputs into their outputs using the corresponding instruction type traces in the specification. A sketch of the algorithm is presented in Algorithm 1. We now present the details in the following.

**Phase I: Searching for candidate trace set.** Due to the large search space, exhaustive search is not practical even for a very short input. Instead, we adopt the idea of training a neural parsing program to explore the search space to find a feasible trace through policy gradient.

---

**Algorithm 1** A sketch of the two-phase reinforcement learning algorithm

1: **function** SEARCH($Net_0$, Lesson, TrainingData)
2:     // Phase 1: nested loop to compute the candidate trace set for each input-output pair
3:     **for** $(input_i, tree_i) \in$ Lesson **do**
4:         $Net_{out} \leftarrow Net_0$
5:         **for** $outItr \leftarrow 1$ to $M_2$ **do**     // Outer loop
6:             Sample an instruction type trace $trace$ using the policy network $Net_{out}$
7:             $Net_{in} \leftarrow Net_{out}$
8:             **for** $InItr \leftarrow 1$ to $M_1$ **do**     // Inner loop
9:                 Execute the programs following $trace$ and
10:                 use the policy network $Net_{in}$ to predict the arguments
11:                 $\hat{T} \leftarrow$ Predicted parse tree
12:                 **if** $diff(\hat{T}, tree_i) = 0$ **then**
13:                     update the candidate trace set for $(input_i, tree_i)$
14:                 **end if**
15:                 Following the REINFORCE algorithm to update $Net_{in}$
16:             **end for**
17:             Following the REINFORCE algorithm to update $Net_{out}$
18:         **end for**
19:     **end for**
20:
21:     // Phase 2: find a satisfiable specification
22:     **for** $(input_i, tree_i) \in$ TrainingData **do**
23:         Assume there are $d$ candidate traces for $(input_i, tree_i)$
24:         Create $\theta_i$ as a $d$-dimensional vector and randomly initialize it
25:     **end for**
26:     **while** True **do**     // It typically terminates within 50 iterations
27:         **for** $(input_i, tree_i) \in$ TrainingData **do**
28:             Sample a candidate trace following the distribution $\textbf{softmax}(\theta_i)$
29:         **end for**
30:         Train a network $Net$ using reinforcement learning as discussed in Section 4.2
31:         **if** Training accuracy is $100\%$ **then**
32:             **return** $Net$
33:         **end if**
34:     **end while**
35: **end function**

---

Specifically, we develop a *two-nested-loop* process to search for the candidate trace set for each input-output pair. In each iteration of the outer loop, we run a forward pass of the model to sample an execution trace including a sequence of instructions and their arguments. We sample the execution trace using the model described in Section 3.2, except that while sampling the next instruction type among valid instruction types, we use the following the distribution instead:

$$p(inst|fid, l, tok) \propto \textbf{softmax}(...)_{inst} + \sigma \tag{5}$$

Here, $\sigma > 0$ is a constant allowing exploration during the search.

After a forward pass, we use the difference between the predicted parse tree and the ground truth as the reward to update the model's parameters predicting the next instruction type using policy gradient. This algorithm is referred to as *learning without supervision on traces*, and we will explain the details in Section 4.2.

If the predicted tree is identical to the ground truth, then we have successfully found a valid instruction type trace, and we add it into the candidate trace set. Otherwise, we test in the inner loops whether the sampled instruction type trace is wrong, or only the arguments are predicted wrongly.

To do so, in the inner loops, we use the sampled instruction type trace in the outer loop as the *candidate* ground truth, and train the model with *weakly supervised learning* method which will be explained in Section 4.2. If any prediction tree during the inner loops matches the candidate

ground truth, we add the sampled instruction type trace to the candidate set. Otherwise, the model's parameters are reverted back to those at the beginning of the inner loop, and the sampled instruction type trace is dropped.

At the end of the outer loop, the candidate trace set is formed, which typically includes 3 to 5 instruction type traces, and the model used during the loop is dropped.

In the description of Phase 1 in Algorithm 1, $M_1$ and $M_2$ are two hyper-parameters, where $M_1$ is the number of iterations for the inner loop, and $M_2$ is the number of iterations for the outer loop. Meanwhile, to escape from a sub-optimal model, we re-initialize the model with the one learned from the previous lesson for every $M_3$ iterations in the outer loop. The values of $M_1$, $M_2$ and $M_3$ for our experiments are described in Appendix E.

**Phase II: Searching for a satisfiable specification.** To find a satisfiable specification, again, the naive idea to perform an exhaustive search requires to explore an exponential number of specifications in the volume of training samples, which is impractical. For example, if each input-output example has 3 candidate traces, the search space of Phase II for a training set of 20 input-output examples has $3^{20} = 3,486,784,401$ instances.

Alternatively, we employ a sampling-based approach. For each input-output pair $(i_k, T_k)$ in the training set, we assume $S_k = \{\mathrm{tr}_{k,1}, ..., \mathrm{tr}_{k,d}\}$ is its candidate trace set including $d$ traces. We sample a trace following the distribution

$$p(\mathrm{tr}_{k,j}) = \mathbf{softmax}(\theta_k)_j \tag{6}$$

where $\theta_k$ is a $d$-dimensional vector. After one trace is sampled for each input-output pair, these traces form a specification, and we try to train a model using the weakly supervised learning algorithm described in Section 4.2 with this specification. If the model can correctly parse all inputs, then we find a satisfiable specification. Otherwise, for each input-output pair $(i_k, T_k)$ that is wrongly parsed, we decrease the probability of sampling current trace in the future by updating $\theta_k$ using:

$$\theta_k \leftarrow \theta_k - \tau \cdot \mathit{diff}(\hat{T}_k, T_k) \cdot \nabla_{\theta_k} \log p(\mathrm{tr}_{k,j}) \tag{7}$$

where $\hat{T}_k$ is the predicted parse tree, and $\tau = 1.0$. We observe that such a sampling-based approach can efficiently sample a satisfiable specification within 30 attempts in our experiments, which could be 8 orders of magnitude smaller than the exhaustive search.

**Curriculum learning.** Searching for a valid trace for a longer input from a randomly initialized model can be very hard. To solve this problem, we use curriculum learning to train the model to learn to parse inputs from shorter length to longer length. In the curriculum, the first lesson contains the shortest inputs. In this case, we randomly initialize the model, and train it to parse all samples in Lesson 1. Afterwards, for each new lesson, we use the parameters learned from the previous lesson to initialize the model. When learning each lesson, all training samples from previous lessons are also added into the training set for the current lesson to avoid catastrophic forgetting (Kirkpatrick et al., 2017). Such a process continues until the model can correctly parse all samples in the curriculum.

## 4.2 Training using Reinforcement Learning

Now we explain how reinforcement learning, especially the policy gradient algorithm REIN-FORCE (Williams, 1992), can be used to update the model during the two-phase training to effectively find a model for both trace exploration and specification satisfiability checking.

In Section 4.1, we explained that we use two versions of the algorithms: *learning with no supervision on traces* explores possible execution traces while the ground truth is not given; and *weakly supervised learning* tries to train the model to fit for a given set of trace specifications. The only difference between the two algorithms is whether a set of ground truth execution traces is given.

In our experiments, we find that the main challenge to apply the REINFORCE algorithm is that the training process is very sensitive to the design of the reward functions. In the following, we first present our design of the reward functions for training the argument prediction sub-networks. In the end, we discuss the different approaches to train the instruction type prediction sub-network when the execution traces are given or not. More details can be found in Appendix D.

**Learning to predict `REDUCE` arguments $n$ and $(c_1, ..., c_m)$.** For the `REDUCE` instruction, our intuition is that if a wrong set of arguments is used, the generated sub-tree will look very different than the ground truth tree. Therefore, we design the reward function based on the difference between the predicted sub-tree and the ground truth.

First, we define the *difference* between two trees $T$ and $T'$, denoted as $diff(T, T')$, to be the edit distance between $T$ and $T'$ (Tai, 1979). Assume $\hat{T}$ is the final generated parse tree and $T_g$ is the ground truth output tree. Our goal is to minimize $diff(\hat{T}, T_g)$, i.e., to 0.

Assume the parse tree constructed by the `REDUCE` instruction is $\hat{T}_r$. Since the final generated parse tree is composed by these smaller trees, a correct parse tree $\hat{T}_r$ should also be a sub-tree of $T_g$. Based on this intuition, we define $mindiff(\hat{T}_r, T_g) = \min_{T \in \mathcal{S}(T_g)} \{diff(\hat{T}_r, T)\}$, where $\mathcal{S}(T_g)$ indicates the set of all sub-trees of $T_g$. If all of the `REDUCE` arguments are predicted correctly, $mindiff(\hat{T}_r, T_g)$ should be 0.

We design the reward function for $n$ and $(c_1, ..., c_m)$ as below:

$$r_{\text{reduce}}(\hat{T}_r) = -\log(\alpha \cdot mindiff(\hat{T}_r, T_g) + \beta)$$

where $\alpha > 1, \beta \in (0, 1)$ are two hyperparameters. In our experiments, we choose $\alpha = 3, \beta = 0.01$.

In addition, we have a more efficient approach to learn the prediction for $n$ via supervised learning. The details can be found in Appendix D.

**Learning to predict `CALL` argument $fid$.** Designing the reward function to learn the prediction of $fid$ is challenging. As we can see in Figure 5, the choice of each $fid$ affects only the prediction of subsequent instruction types. Our design of the reward function for $fid$ takes this into account. Intuitively, a wrong guess of $fid$ will result in incorrect subsequent predicted instruction types. Based on this intuition, we design the reward function as follows:

$$r_f(fid^{(t)}) = \sum_{j=t+1}^{t'} \log p(\hat{inst}^{(j)} | fid^{(j)}, l^{(j)}, tok^{(j)})_{inst^{(j)}}$$

where $t$ indicates the current step to execute a `CALL` instruction, $t'$ the next step to execute a `CALL` instruction, $\hat{inst}^{(j)}$ and $inst^{(j)}$ the predicted and ground truth instruction types, and $(fid^{(j)}, l^{(j)}), tok^{(j)}$ the frame at the stack top and the next input token at step $j$. Basically, the reward function $r_f$ accumulates the negation of the cross-entropy loss of the predicted instructions from the current `CALL` instruction till the next one. More explanations can be found in Appendix D.

**Learning to predict the next instruction type.** When the execution traces are given, training the next instruction prediction sub-network is a supervised learning problem, and can be solved using NPI-style training approaches (Reed & De Freitas, 2016; Cai et al., 2017).

When the execution traces are not given, we use REINFORCE to update the sub-network for next instruction type prediction as well. In particular, assume the prediction tree generated during the forward pass is $\hat{T}$ and the ground truth of the output is $T_g$. Then the reward function is design to be

$$-\log(\alpha \cdot diff(\hat{T}, T_g) + \beta). \tag{8}$$

## 5 A RUNNING EXAMPLE

In this section, we detail the design of our reinforcement learning-based algorithm using a running example. For illustration purposes, we use a toy language, whose grammar is still challenging to learn. The grammar is very simple: all expressions composed by addition and multiplication. We allow `x` and `y` as identifiers and `0` and `1` for literals. We refer to this language as Addition-Multiplication (AM), and it is also a small subset of the WHILE language, which we will evaluate in the next section. The curriculum of AM language is presented below.

$$
\begin{array}{cccccc}
x + y & x * y & x + 0 & x * 0 & 0 + 1 & 0 * 1 \\
y + x + 0 & y + 0 + x & 0 + x + y & y * x * 0 & y * 0 * x & 0 * x * y \\
y * x + 0 & y + x * 0 & 0 * 1 + x & 0 + 1 * x & y + 1 + x + 0 & y + 1 + x * 0 \\
y + 1 * x + 0 & y + 1 * x * 0 & y * 1 + x + 0 & y * 1 + x * 0 & y * 1 * x + 0 & y * 1 * x * 0
\end{array}
$$

**The difficulty of the problem: search space.**    To understand the difficulty of the problem, we would like to emphasize that learning a correct parser is equivalent to finding the correct execution trace for each input-output example. On the one hand, if the correct parser is learned, then we can recover the correct trace for each input-output example by simply executing the parser over the input; on the other hand, if the correct trace of each input-output example is recovered, then we can easily use supervised learning to train the parsing program. Therefore, the difficulty of the learning problem can be measured by the volume of the search space.

We now estimate the size of the entire search space. For AM grammar, we parameterize the LL machine with $K = 3$ and $F = 3$, and there are $4$ different non-terminals. We present the number of the shortest valid execution traces versus the input length below.

| Input length | Length of correct exec. traces | Number of shortest valid exec. traces |
|---|---|---|
| 3 | 9 | $1,572$ |
| 5 | 15 | $2,771,712$ |
| 7 | 21 | $7,458,826,752$ |

Here, we call an execution trace to be *valid* if the trace ends with a valid `FINAL` instruction, but the output tree does not necessarily match the ground truth. Meanwhile, we only calculate the number of traces that are of the same lengths as the correct execution traces, where *correct* traces mean the shortest traces that can lead to the ground truth output trees. We present the lengths of the shortest correct traces above, and denote traces of the same lengths as *shortest* traces for brevity.

Given our training curriculum, a naive way to find a correct set of execution traces for all input-output examples is to perform an exhaustive search over the space of $1572^6 \times 2771712^{10} \times 7458826752^8 = 3.87 \times 10^{162}$. Such a huge search space makes the exhaustive search approach impractical. Notice that the number of valid execution traces increases exponentially as the input length increases.

Meanwhile, this search space is just for a simple grammar, i.e., AM. For more complex languages that we will use in our evaluation, i.e., WHILE and LAMBDA, the value of $F$ and the number of different non-terminals are larger. Thus, the total number of valid execution traces for WHILE or LAMBDA is much higher than for AM. Also, the average input lengths for WHILE and LAMBDA are 9.3 and 5.6 respectively, which further increases the search space volume significantly.

**Motivating the two-phase algorithm.**    Given the huge search space, we design our techniques to reduce the search space. We observe several aspects why the search space is huge. The first issue is that the large search space is mainly due to the *rule of product* when considering the combinations of traces for all input-output examples. When considering only one input-output example, e.g., an input of length 7, the size of the search space is around 7.5 billion. Though it is still large, such a size is more tractable than $3.87 \times 10^{162}$. This inspires the idea of two-phase learning: the first phase searches for correct traces for each input-output example, and the second phase searches a combination of traces for different samples. In doing so, the first phase only needs to perform a search over a space of $6 \times 1572 + 10 \times 2771712 + 8 \times 7458826752 = 5.97 \times 10^{10}$ instances, which is still large, but much more amenable than the original search space.

Having done this separation, we can focus on the rest issues: (1) $5.97 \times 10^{10}$ is still a large search space, and thus performing the search in the first phase is not efficient enough yet; and (2) how to make the second phase efficient and effective is unclear. In the following, we will explain how our design resolves these issues.

**Reducing the search space: using instruction type traces instead of execution traces.**    We observe that the total number of the shortest valid execution traces for an input-output example in the curriculum can be as large as $7.5 \times 10^9$, which is still large for an exhaustive enumeration. Notice that most of them are equivalent to each other up to permutation of the instruction arguments. In fact,

the total number of the shortest valid *instruction type traces* for each example is much smaller, as we show below.

| Input length | Number of shortest valid exec. traces | Number of shortest valid inst. type traces |
|---|---|---|
| 3 | $1,572$ | 9 |
| 5 | $2,771,712$ | 382 |
| 7 | $7,458,826,752$ | $23,816$ |

Therefore, enumerating valid instruction type traces can be more efficient than enumerating valid execution traces. This observation inspires the nested-loop algorithm in the first phase. In fact, the outer loop uses reinforcement learning to enumerate different instruction type traces, while the inner loop verifies whether an instruction type trace can be instantiated as an execution trace by searching for arguments using reinforcement learning. We will defer more discussion about the inner loop later.

Meanwhile, using instruction type traces also helps to reduce the search space of the second phase. For example, among all valid instruction type traces, our training algorithm typically finds 3 to 5 traces that can lead to the correct output for each input-output example. Thus, the entire search space is of the order of $3^n$ to $5^n$, where $n$ is the total number of input-output examples being searched together. On the other hand, if all arguments are counted as well, then for each input-output examples, these instruction type traces correspond to tens of execution traces leading to the correct output. In this case, the search space is orders of magnitude larger than considering only instruction type traces. Therefore, using instruction type traces is also a key to reduce the size of the search space in the second phase.

**Reinforcement learning-based sampling versus enumeration.** In the first phase, we choose to use reinforcement learning to explore different possibilities of instruction type traces instead of exhaustive enumeration. This design is based on the following observation. Although we observe that the total number of valid instruction type traces is small for short inputs, this number can grow exponentially large when the input length increases. For example, for the WHILE language, a majority of the training inputs have a length larger than 9. In such a case, exhaustively searching over all valid instruction type traces is not efficient, and using a sampling based approach is typically more effective. This observation is consistent with previous work (Bergstra & Bengio, 2012; Gulwani et al., 2017). Note that our design of the search process in the second phase is also inspired by the same observation.

There is a caveat for both strategies: it may not be easy to know the length of the shortest correct instruction type traces in advance. In the exhaustive search approach, one can enumerate each length of traces in the ascending order and check if there exists a correct instruction type trace. Using a reinforcement learning approach, this process could be reversed: since each outer loop may sample an arbitrarily long instruction type trace, it may sample several longer instruction type traces before reaching the ones with the minimal length. This may cause our reinforcement learning algorithm to run longer for a short input; but for a long input, the sampling approach typically can find the set of candidate instruction type traces much sooner than using the exhaustive enumeration approach. In our evaluation, we observe that typically setting $M_2 = 10,000$ is sufficiently large to find a good set of candidate traces regardless of the input length.

There is a side benefit of using a reinforcement learning-based sampling in the outer loop: the same RL framework can be used in the inner loop to search for a valid execution trace instantiation from the instruction type trace. Thus, the RL algorithm can also be used as an efficient verification tool to check whether a valid instruction type trace is correct or not.

**The effectiveness of training curriculum.** The training curriculum helps with the training in three aspects. First, the RL algorithm has a caveat: for a long input, RL algorithm cannot find even one correct instruction type trace from a cold start. Thus, curriculum learning can help to mitigate this issue. In particular, when searching for correct instruction type traces for an example in one lesson, the model is initialized with parameters that can fit to all examples in previous lessons. In doing so, the RL algorithm can effectively skip many obviously bad traces, and thus find the correct ones more efficiently.

Second, the training curriculum can also help RL to skip those instruction type traces that are correct for the examined input-output examples, but are inconsistent with other examples in previous lessons. For our AM language, we provide the number of correct instruction traces as well as the number of

| Example | # of correct traces | Example | # of correct traces | Example | # of correct traces |
|---|---|---|---|---|---|
| x + y | 9 (3) | x * y | 9 (3) | x + 0 | 9 (3) |
| x * 0 | 9 (3) | 0 + 1 | 9 (3) | 0 * 1 | 9 (3) |
| y + x + 0 | 99 (11) | y + 0 + x | 99 (11) | 0 + x + y | 99 (11) |
| y * x * 0 | 99 (11) | y * 0 * x | 99 (11) | 0 * x * y | 99 (11) |
| y * x + 0 | 99 (11) | y + x * 0 | 81 (9) | 0 * 1 + x | 99 (11) |
| 0 + 1 * x | 81 (9) | y + 1 + x + 0 | 1107 (41) | y + 1 + x * 0 | 891 (33) |
| y + 1 * x + 0 | 1053 (39) | y + 1 * x * 0 | 891 (33) | y * 1 + x + 0 | 1107 (41) |
| y * 1 + x * 0 | 891 (33) | y * 1 * x + 0 | 1107 (41) | y * 1 * x * 0 | 1107 (41) |

Table 2: The numbers of correct instruction traces and instruction type traces for each example in the AM training set. The two numbers are provided in the column of "# of correct traces" in the form of $n(m)$, where $n$ (outside the brackets) indicates the number of instruction traces, and $m$ (inside the brackets) indicates the number of instruction type traces.

correct instruction type traces for each example in Table 2. From the table, we can observe that the number of correct traces increases significantly with respect to the increase of the input length. Using curriculum training, we can reduce the number of candidate instruction type traces to be 3 to 5 for each example regardless of the input length, which thus further reduces the search space for Phase II.

Third, it can further reduce the search space of the second phase. Assume the algorithm finds 3 candidate instruction type traces for each input-output pair. Since there are 24 examples in the training set, the search space of the second phase can be as large as $3^{24} = 2.82 \times 10^{11}$. When following the training curriculum, each lesson may have only 6 examples. Thus, the search space for each lesson in the second phase can be as few as $3^6 = 729$, i.e., 8 orders of magnitude smaller. When training on the next lesson, since the instruction type traces for examples in previous lessons have been determined, the search can focus on the current lesson. Therefore, using a training curriculum can further reduce the search space while guaranteeing that the model is not overfitting to a subset of examples.

## 6 EVALUATION

To show that our approach is general and able to learn to parse different types of context-free languages using the same architecture and approach, we evaluate our approach on two tasks to learn a parser for an imperative language WHILE and an ML-style (Milner, 1997) functional language LAMBDA respectively. WHILE and LAMBDA contain 73 and 66 production rules, and their parsing programs can be implemented in 89 and 46 lines of Python code respectively. Notice that these programs are more sophisticated than previous studied examples. For example, Quicksort studied in (Cai et al., 2017) can be implemented in 3 lines of Python code, and FlashFill tasks studied in (Devlin et al., 2017; Parisotto et al., 2017) can be implemented in 10 lines of code in their DSL. Grammar specifications of the two languages are presented in Appendix G and H respectively.

For each task, we prepare three training sets: (1) **Curriculum:** a well-designed training curriculum including 100 to 150 examples that enumerates all language constructors; (2) **Std-10:** a training set includes all examples in the curriculum, and also 10,000 additional randomly generated inputs with length 10 on average; and (3) **Std-50:** a training set includes all examples in the curriculum, and also 1,000,000 additional randomly generated inputs with length 50 on average. In all datasets, all ground truth parse trees are provided. Note that once our model learns to parse all inputs in the curriculum, it can parse all inputs in training sets (2) and (3) for free. We include two standard training sets, i.e., Std-10 and Std-50, to allow a fair comparison against baseline approaches, which typically require a large amount of training data.

We compare our approach with two sets of baselines. The first class of approaches learn end-to-end neural network models as the program. This class includes a sequence-to-sequence approach (seq2seq) (Vinyals et al., 2015b), a sequence-to-tree (seq2tree) approach (Dong & Lapata, 2016), and LSTM with unbounded memory (Grefenstette et al., 2015). In particular, we evaluate all three variants proposed in (Grefenstette et al., 2015). The second class includes the state-of-the-art approach for neural program synthesis, i.e., RobustFill (Devlin et al., 2017), which learns a discrete program in a DSL.

While-Lang

| Train | Test | Ours | Seq2seq | Seq2tree | Stack LSTM | Queue LSTM | DeQue LSTM | Robust-Fill (Projected) |
|---|---|---|---|---|---|---|---|---|
| Curriculum | Training | **100%** | 81.29% | **100%** | **100%** | **100%** | **100%** | 13.67% |
| | Test-10 | **100%** | 0% | 0.8% | 0% | 0% | 0% | 0% |
| | Test-100 | **100%** | 0% | 0% | 0% | 0% | 0% | 0% |
| | Test-1000 | **100%** | 0% | 0% | 0% | 0% | 0% | 0% |
| | Test-5000 | **100%** | 0% | 0% | 0% | 0% | 0% | 0% |
| Std-10 | Training | **100%** | 94.67% | **100%** | 81.01% | 72.98% | 82.59% | 0.19% |
| | Test-10 | **100%** | 20.9% | 88.7% | 2.2% | 0.7% | 2.8% | 0% |
| | Test-100 | **100%** | 0% | 0% | 0% | 0% | 0% | 0% |
| | Test-1000 | **100%** | 0% | 0% | 0% | 0% | 0% | 0% |
| Std-50 | Training | **100%** | 87.03% | **100%** | 0% | 0% | 0% | 0.0019% |
| | Test-50 | **100%** | 86.6% | 99.6% | 0% | 0% | 0% | 0% |
| | Test-500 | **100%** | 0% | 0% | 0% | 0% | 0% | 0% |
| | Test-5000 | **100%** | 0% | 0% | 0% | 0% | 0% | 0% |

Lambda-Lang

| Train | Test | Ours | Seq2seq | Seq2tree | Stack LSTM | Queue LSTM | DeQue LSTM | Robust-Fill (Projected) |
|---|---|---|---|---|---|---|---|---|
| Curriculum | Training | **100%** | 96.47% | **100%** | **100%** | **100%** | **100%** | 29.21% |
| | Test-10 | **100%** | 0% | 0% | 0% | 0% | 0% | 0% |
| | Test-100 | **100%** | 0% | 0% | 0% | 0% | 0% | 0% |
| | Test-1000 | **100%** | 0% | 0% | 0% | 0% | 0% | 0% |
| | Test-5000 | **100%** | 0% | 0% | 0% | 0% | 0% | 0% |
| Std-10 | Training | **100%** | 93.53% | **100%** | 0% | 95.93% | 2.23% | 0.26% |
| | Test-10 | **100%** | 86.7% | 99.6% | 0% | 6.5% | 0.1% | 0% |
| | Test-100 | **100%** | 0% | 0% | 0% | 0% | 0% | 0% |
| | Test-1000 | **100%** | 0% | 0% | 0% | 0% | 0% | 0% |
| Std-50 | Training | **100%** | 66.65% | 89.65% | 0% | 0% | 0% | 0.0026% |
| | Test-50 | **100%** | 66.6% | 88.1% | 0% | 0% | 0% | 0% |
| | Test-500 | **100%** | 0% | 0% | 0% | 0% | 0% | 0% |
| | Test-5000 | **100%** | 0% | 0% | 0% | 0% | 0% | 0% |

Table 3: Experimental results on While-Lang and Lambda-Lang dataset. We evaluate our approach (ours), seq2seq (Vinyals et al., 2015b), seq2tree (Dong & Lapata, 2016), with Stack LSTM, Queue LSTM and DeQue LSTM from (Grefenstette et al., 2015). "Std-10" indicates the standard training set with 10,000 samples of length 10, "Std-50" indicates the standard training set with 1,000,000 samples of length 50, and "Curriculum" indicates the specially designed learning curriculum. "Test-LEN" indicates the testset including inputs of length LEN.

For testing, we create six levels of testsets, i.e., Test-10, Test-50, Test-100, Test-500, Test-1000 and Test-5000, where each input has 10, 50, 100, 500, 1000 and 5000 tokens on average respectively. Each test set contains 1000 randomly generated expressions. We guarantee that test data does not overlap with training samples. Table 3 shows experimental results on WHILE and LAMBDA languages. We discuss the results below.

**Observations on our approach.** We observe that once our neural parsing program is trained to achieve 100% accuracy on the training data, it can always achieve 100% test accuracy on arbitrary test samples regardless of their lengths.

We emphasize that all the results of our approach are achieved by using the curriculum learning approach. Without curriculum training, our approach cannot train any effective models, and the test accuracy is 0% when the test input is longer than training inputs. Also, we observe that our two-phase training algorithm can always make the neural network be trained to achieve 100% on the

curriculum training set. Further, in our evaluation, we observe that the training curriculum is very important to regularize the reinforcement learning process.

Therefore, our evaluation demonstrates that the combination of our three ideas enables us to learn a program to achieve $100\%$ accuracy on test samples that can be even $500\times$ longer than the training ones, while baseline approaches are hard to even achieve a test accuracy that is greater than $0\%$.

**Observations on approaches to learn end-to-end neural network models.** We first observe that when the length of test samples is larger than training ones, the test accuracy drops to 0% regardless of the end-to-end differential approach being evaluated. This illustrates that none of these approaches can generalize to longer inputs. As we have explained, it is very hard to enforce the learned model to always represent a parser to handle arbitrarily long inputs. Thus even the well-trained models simply overfit to the training samples, and cannot generalize to longer inputs.

Also, when training on the curriculum dataset, no approaches can generalize to any test data. This is simply because the training set contains too few instances, so that the overfitting phenomenon explained above becomes more prominent.

When test samples are of the same length as training ones and training with the standard training sets, we observe that seq2tree performs better than seq2seq. We attribute this phenomenon to the reason that the seq2tree model essentially employs the recursion idea in its decoder design. In fact, in the decoding phase, different from seq2seq, which generates the entire sequence at once, seq2tree traverses down along the paths from the root to leaves and recursively decodes each layer of the parse tree along a path. On WHILE and LAMBDA tasks, it can even achieve an accuracy of almost $100\%$ on Test-50 and Test-10 respectively, showing that this recursive decoding approach is effective. However, the encoding phase of seq2tree is not recursive. This hinders its generalization to longer inputs.

Meanwhile, for both seq2seq and seq2tree models, when they are trained on Std-50 training set, the gap between the training accuracy and the test accuracy is much smaller than the ones trained on Std-10 training set. For example, on WHILE dataset, the test accuracy of seq2seq is around $20\%$ on Test-10; however, when trained on Std-50, it can reach an accuracy of above $85\%$ on Test-50. This observation suggests that the models overfit to the Std-10 training set. When Std-50 is used for training, on the other hand, the overfitting issue is mitigated. In addition, we notice that while the test accuracy for the WHILE task increases when Std-50 is used for training, it drops for the LAMBDA task. We attribute it to the fact that the size of the parse tree in LAMBDA language is much larger than the parse tree in WHILE language given the input with the same number of tokens. Thus, when the inputs get longer, the model is harder to fit to the parse trees of larger sizes.

We also observe that models proposed in (Grefenstette et al., 2015) perform poorly, and much worse than the other two end-to-end neural network approaches. Grefenstette et al. (2015) proposes LSTM decoders with unbounded memory to generalize the idea of neural pushdown automaton, which was designed to handle the parsing tasks. Our results show that when these models are trained on Std-10, they suffer severely from overfitting. Further, when trained on Std-50, the training accuracy drops to 0%. We attribute the poor performance to the fact that the production and transduction rules developed in the two evaluated languages are too complex for such architectures to learn effectively. In fact, Grefenstette et al. (2015) reports that all three proposed approaches perform poorly on the bi-gram flipping task, which is a much simpler sub-task of the two languages in consideration. This further illustrates the challenges of training such end-to-end differentiable neural networks to simulate even simple data structures such as stacks or queues.

**Observations on approaches to synthesize discrete programs.** Note that the training of Robust-Fill, as described in (Devlin et al., 2017), do not use the input-output examples in the training set, but construct their own training set instead. The source code of RobustFill is not available, and our re-implementation cannot produce any meaningful programs.

To make a fair comparison, we compute a *projected* accuracy which is the accuracy that can be achieved by the most effective program in the space of the RobustFill DSL. Note that given RobustFill can produce programs of length of up to 10, the entire output program space of RobustFill is finite, though exponentially large. However, we notice that we can efficiently enumerate the small sub-set

of effective programs using a simple heuristic to cut ineffective constructors in a program. We detail the algorithm in Appendix K.

The results in Table 3 report an upper bound of the accuracy that can be achieved by the RobustFill approach. We can observe that the best program in the RobustFill DSL space can only fit to a small subset of the training data, and cannot generalize to longer test inputs due to the length of the programs is limited. Essentially, the outputs of a program in the RobustFill DSL space are bounded by the length of the program itself (see Appendix K for a discussion), while the outputs of our parsing problem can grow arbitrarily long. When the test samples become longer, the RobustFill approach will soon fail with 0% accuracy.

Therefore, to make the RobustFill approach able to handle our parsing task, it is necessary to develop a novel DSL with enough expressiveness (i.e., supporting recursion to allow arbitrarily long outputs). However, this is a highly non-trivial task, and is out of the scope of this work.

# 7 RELATED WORK

We now present a high-level overview of related work. A more in-depth discussion about the relationship between our work and previous work is presented in Appendix A.

Recent works propose to use sequence-to-sequence models (Vinyals et al., 2015b; Aharoni & Goldberg, 2017) and their variants (Dong & Lapata, 2016) to directly generate parse trees from inputs. However, they often do not generalize well, and our experiments show that their test accuracy is almost 0% on inputs longer than those seen in training.

Other works study learning a neural program to operate a *Shift-Reduce* machine (Andor et al., 2016; Chen & Manning, 2014; Yogatama et al., 2016) or a top-down parser (Dyer et al., 2016) to perform parsing tasks for natural languages. In these works, the execution traces are easy to recover from input-output pairs, while in our work the traces are hard to recover.

Recent works study learning neural programs and differentiable machines (Graves et al., 2014; Kurach et al., 2015; Joulin & Mikolov, 2015; Kaiser & Sutskever, 2015; Bunel et al., 2016). Their proposed approaches either do not generalize to longer inputs than those seen during training, or are evaluated only on simple tasks. In particular, StackRNN (Joulin & Mikolov, 2015) also studies learning context-free languages, but their main focus is to generate language instances, while our goal is to learn the parser. Employing a similar idea, Grefenstette et al. (2015) design an end-to-end differentiable push-down automaton for transduction tasks, which are similar to ours. As we will demonstrate, such an approach has even worse generalization than a sequence-to-sequence model.

On the other hand, other works study neural programs operating non-differentiable machines (Cai et al., 2017; Li et al., 2017; Reed & De Freitas, 2016; Zaremba et al., 2016; Zaremba & Sutskever, 2015), but in these works, either extra supervision on execution traces is needed during training (Reed & De Freitas, 2016; Cai et al., 2017; Li et al., 2017), or the trained model cannot generalize well (Zaremba et al., 2016; Zaremba & Sutskever, 2015). In particular, Zaremba et al. (2016) study learning simple algorithms from input-output examples; however, the approach fails to generalize on very simple tasks, such as 3-number addition. Our work is the first one demonstrating that a neural program achieving full generalization to longer inputs can be trained from input-output pairs only.

Another line of research studies using neural networks to synthesize a program in a domain-specific language (DSL). Recent works (Devlin et al., 2017; Parisotto et al., 2017) study using neural networks to generate a program in a DSL from a few input-output examples for the FlashFill problem (Gulwani et al., 2012; Gulwani, 2011). However, the DSL contains only simple string operations, which is not expressive enough to implement a parser. Meanwhile, in these works, they can only successfully synthesize programs with lengths not larger than 10. These constraints make their approaches unsuitable for our problem currently. DeepCoder (Balog et al., 2017) presents a neural network-based search technique to accelerate search-based program synthesis. Again, lengths of the synthesized programs in this work are at most 5, while the parsing program that we study in this work is much more complex. There are other approaches (Ellis et al., 2016) that employ SMT solvers to sample programs. Again, it is only demonstrated to solve a subset of the FlashFill problem and several simple array manipulation tasks.

## 8    CONCLUSION AND FUTURE WORK

In this work, we move a significant step forward to learn complex programs from input-output examples only. In particular, we propose a novel class of grammar induction problems to learn a parser from the input-tree pairs. We demonstrate that the parsing problems are more challenging as most existing approaches fail to generalize, i.e., the test accuracy is $0\%$. To solve this problem, we reveal three novel challenges and propose novel principled approaches to tackle them. First, we promote a hybrid approach to learn a neural program operating a non-differentiable machine to effectively restrict the learned programs within the space of interest. Second, we design the machine to bake-in the notion of recursion to make the learned neural programs generalizable. Third, we propose a novel two-phase reinforcement learning-based algorithm to effectively train such a neural program. Combining the three techniques, we demonstrate that the parsing problem can be fully solved on two diverse instances of grammars.

In the future, we are interested in both the domain of parsing problems and even more complex programs. For the parsing problems, we are interested in recovering the production rules from input-output examples, rather than only learning the parser, and relax several technical assumptions, such as the knowledge of the terminal set and hyper-parameter $K$, which is the maximum number elements in the list of each stack frame. For more complex programs, we are interested in extending our approach to learn algorithms on more complex data structures such as trees and graphs.

### ACKNOWLEDGEMENT

We thank Richard Shin, Dengyong Zhou, Yuandong Tian, He He, Yu Zhang, and Warren He for their helpful discussions. This material is in part based upon work supported by the National Science Foundation under Grant No. TWC-1409915, Berkeley DeepDrive, and DARPA STAC under Grant No. FA8750-15-2-0104. Any opinions, findings, and conclusions or recommendations expressed in this material are those of the author(s) and do not necessarily reflect the views of the National Science Foundation.

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

## A    MORE DISCUSSION ABOUT RELATED WORK

**Grammar induction.**    Learning the grammar from a corpus of examples has long been studied in the literature as the *grammar induction* problem, and algorithms such as L-Star (Angluin, 1987) and RPNI (Oncina & García, 1992) have been proposed to handle regular expressions. In contrast, in this work, we are interested in learning context-free languages (Chomsky, 1956), which is much more challenging than learning regular languages (De la Higuera, 2010).

**Sequence-to-sequence style approaches.**    Recent works propose to use sequence-to-sequence models (Vinyals et al., 2015b; Aharoni & Goldberg, 2017) and their variants (Dong & Lapata, 2016) to directly generate parse trees from inputs. However, they often do not generalize well, and our experiments show that their test accuracy is almost 0% on inputs longer than those seen in training.

**Parsing approaches using machines in NLP literatures.**    A recent line of research (Andor et al., 2016; Chen & Manning, 2014; Yogatama et al., 2016) studying dependency parsing employs neural networks to operate a *Shift-Reduce* machine. However, each node in the generated dependency tree corresponds to an input token, while in our problem, there is not a direct correspondence between the internal nodes in parse trees and the input tokens. Further, RNNG (Dyer et al., 2016) learns a neural program operating a top-down parser to generate parse trees, which include non-terminals. However, as explicitly stated in the paper (Dyer et al., 2016), the input tokens align well with the pre-order traversal of the parse tree. In our work, such order is often not preserved and the correspondence is hard to be recovered. Thus, these approaches do not directly apply to our problem.

**Neural program induction.**    Recent works study learning neural programs and differentiable machines (Graves et al., 2014; Kurach et al., 2015; Joulin & Mikolov, 2015; Kaiser & Sutskever, 2015; Bunel et al., 2016). Their proposed approaches either do not generalize to longer inputs than those seen during training, or are evaluated only on simple tasks. In particular, StackRNN (Joulin & Mikolov, 2015) also studies learning context-free languages, but their main focus is to generate language instances, while our goal is to learn the parser. Grefenstette et al. (2015) adopts a similar idea for learning to transduce. However, as we demonstrate, such an approach performs poorly and even worse than a sequence-to-sequence model.

On the other hand, other works study neural programs operating non-differentiable machines (Cai et al., 2017; Li et al., 2017; Reed & De Freitas, 2016; Zaremba et al., 2016; Zaremba & Sutskever, 2015), but in these works, either extra supervision on execution traces is needed during training (Reed & De Freitas, 2016; Cai et al., 2017; Li et al., 2017), or the trained model cannot generalize well (Zaremba et al., 2016; Zaremba & Sutskever, 2015). In particular, (Zaremba et al., 2016) studies learning simple algorithms from input-output examples; however, the approach fails to generalize on very simple tasks, such as 3-number addition. Our work is the first one demonstrating that a neural program achieving full generalization to longer inputs can be trained from input-output pairs only.

**Neural program synthesis.**    Another line of research studies using neural networks to synthesize a program in a domain-specific language (DSL). Recent works (Devlin et al., 2017; Parisotto et al., 2017) study using neural networks to generate a program in a DSL from a few input-output examples for the FlashFill problem (Gulwani et al., 2012; Gulwani, 2011). However, the DSL contains only simple string operations, which is not expressive enough to implement a parser. Meanwhile, in these works, they can only successfully synthesize programs with lengths not larger than 10. These constraints make their approaches unsuitable for our problem currently. DeepCoder (Balog et al., 2017) presents a neural network-based search technique to accelerate search-based program synthesis. Again, lengths of the synthesized programs in this work are at most 5, while the parsing program that we study in this work is much more complex. There are other approaches (Ellis et al., 2016) that employ SMT solvers to sample programs. Again, it is only demonstrated to solve a subset of the FlashFill problem and several simple array manipulation tasks. As all these approaches follow the same paradigm to synthesize a program in the DSL used in RobustFill, in Appendix K, we give a fundamental reason why such approaches cannot generalize to handle the parsing problems.

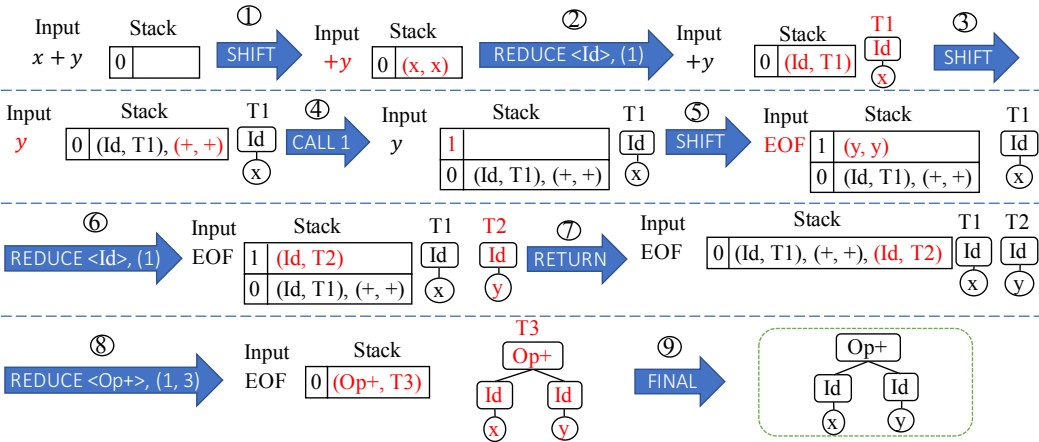

Figure 4: A copy of Figure 2 to provide a full example to parse `x+y` into a parse tree. 9 instructions are executed to construct the final tree. We use red labels to illustrate the changes after performing each operation. For ease of illustration, if a tree has only one node, which is a terminal, then we simply use this terminal to represent both the root node and the tree in the stack; otherwise, we draw the tree next to the stack, and refer to it with a unique label in the stack.

## B  LL MACHINES

We start with the presentation of the LL machines' design. It is inspired by the $LL(1)$ parsing algorithm (Parr & Quong, 1995), although we do not require the readers to be familiar with the $LL(1)$ algorithm. Throughout the description, we use Figure 2 (we provide a copy in Figure 4 which is closer to the description below) as a running example to illustrate the concepts.

**States.**   An LL machine maintains a sequence of (partial) input tokens and a stack of *frames* as its internal state. Each stack frame is an ID-list pair, where the ID is a *function ID*, which will be explained later, and in the list are $(n, T)$ pairs, where $T$ is a parse tree, and $n$ is the root node of $T$. For example, in Figure 4, after step 6, the stack frame at the top contains an ID 1 and a list of one element $(\texttt{Id}, \text{T2})$.

**Instructions.**   An LL machine has five types of instructions: SHIFT, CALL, RETURN, REDUCE, and FINAL. A parser operates an LL machine using these five types of instructions to construct the parse tree recursively. In the following, we explain these instructions and how they are used for parsing an input. To begin with, the stack contains one frame $(0, [])$, where $[]$ denotes an empty list.

A SHIFT instruction (e.g., steps 1, 3, and 5 in Figure 4) removes the next token $t$ from the input sequence, constructs a one-node tree $T$ consisting of $t$, and appends $(t, T)$ to the end of the stack top's list. The SHIFT instruction has no argument.

When the parser tries to parse a sub-expression as a sub-tree, it uses a CALL instruction to create a new stack frame. For example, before step 4, the sub-expression "y" needs to be parsed into T2 with root Id. In this case, a CALL instruction is executed to push a new frame with an empty list onto the stack. CALL has an argument $fid$, which is the function ID of the new frame at the stack top. This function ID carries information from the previous frame to the new one, e.g., to help decide the boundary of the sub-expression. In Figure 5, for example, when parsing "x+y⋆z" and "x⋆y⋆z", once the first two tokens (i.e., "x+" and "x⋆") are consumed, the parser executes a CALL instruction to create a new frame to parse the sub-expressions "y⋆z" and "y" respectively. Since the remaining input sequences (i.e., "y⋆z") are the same in both cases, the function IDs provide the only clue to detect the boundaries of the sub-expressions.

The parser issues a REDUCE instruction to construct a larger tree, once all children of its root are constructed and laid out in the top frame's list. REDUCE $n, (c_1, ..., c_m)$ has two arguments for specifying how to construct the new tree. The root of the newly constructed tree is $n$ and has $m$

children. The $j$-th child of $n$ is the $c_j$-th tree in the stack top's list. For example, in Figure 4, after step 8, T1 and T2 are combined to construct T3. The list in the top frame contains three elements, i.e., $(\text{Id}, \text{T1})$, $(+, +)$, and $(\text{Id}, \text{T2})$. In this case, the REDUCE argument $n$ is Op+, indicating that T3's root is Op+; for the second argument $(c_1, ..., c_m)$, $m = 2$, $c_1 = 1$ and $c_2 = 3$, indicating that the first and third elements in the list (i.e., T1 and T2) constitute the first and second children of T3. Note that the children of the root are ordered.

After a sub-expression is converted into a tree using the REDUCE instruction, a RETURN instruction can be executed to move the tree into the previous stack frame, so that it can be used to further construct larger trees. Formally, when the list in the top frame contains only one element $(n, T)$, RETURN (e.g., step 7 in Figure 4) pops the stack, and appends $(n, T)$ to the end of new stack top's list.

When all input tokens are consumed and the stack contains only one tree, the parser executes FINAL (e.g., step 9 in Figure 4) to terminate the machine. Both RETURN and FINAL have no arguments.

**Valid instruction set.**    At each step, an LL machine provides a set of *valid instructions* that can be executed. In doing so, the machine can guarantee that the state remains valid if the instructions to be executed are always chosen from this set.

We now demonstrate that how our LL machines restrict the space of the learned programs. To achieve this, we impose several constraints on the instruction types that can be applied at each timestep. We denote the current stack top as $(\mathit{fid}, l)$, the length of $l$ as L, and the first token of the current input as $tok$ ($tok = \text{EOF}$ if the current input is empty). Meanwhile, we assume that each stack frame's list has at most $K$ elements, and we will explain why this assumption holds later. The constraints for when each of the five instructions is allowed are as below:

1. SHIFT: it is allowed if $tok \neq \text{EOF}$ and $L < K$.
2. CALL: it is allowed if $tok \neq \text{EOF}$, $0 < L < K$, and the instruction type at previous timestep is not CALL. For its argument $\mathit{fid}'$, $0 \leq \mathit{fid}' < F$, where $F > 0$ is a hyper-parameter indicating the number of different function IDs.
3. RETURN: it is allowed if the current stack has more than one frame, and $L = 1$.
4. REDUCE: it is allowed if $L > 0$, and the instruction type at previous timestep is not REDUCE. For REDUCE arguments $n$ and $(c_1, ..., c_m)$, $n$ is chosen from the non-terminal set, and $1 \leq c_i \leq L$ for $1 \leq i \leq m$.
5. FINAL: it is allowed if $tok = \text{EOF}$, $L = 1$, and the current stack has only one frame.

**More details.**    Then we explain why we can safely assume that there exists $K$ such that each stack frame's list has at most $K$ elements. As the parsing program continues, each stack frame's list contains partially finished sub-trees that correspond to a prefix of one production rule in the grammar. Since the length of production rules in a context-free grammar is finite, we can assume that the upper bound of the length is $K$. According to the instruction constraints imposed by LL machines, using the same $K$ as the upper bound on the length of each stack frame's list, we can ensure that for each

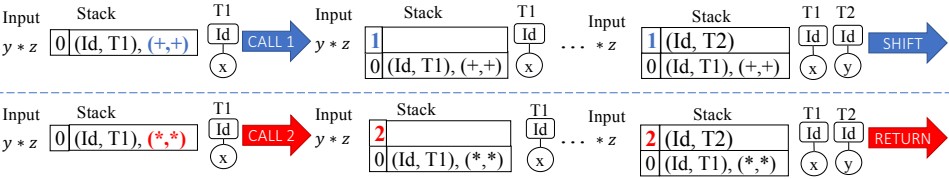

Figure 5: The (partial) execution traces for parsing "x+y*z" (above) and "x*y*z" (bottom) respectively. For "x+y*z", "y*z" needs to be associated; when "y" is reduced to a tree with the root Id, the token "*" needs to be shifted into the stack top. On the other hand, for "x*y*z", "x*y" needs to be associated, thus after "y" is reduced, the parse tree of "x*y" should be popped before shifting the next token "*" in the input. In this case, only the function ID in the stack top, i.e., 1 (above) or 2 (bottom), can distinguish whether SHIFT or RETURN should be executed next.

input in the grammar, there exists a trace satisfying such constraints that can parse the input to its parse tree correctly.

## C   MODEL ARCHITECTURE

We explain how the model chooses the instruction to be executed at each step. As for the prediction of instruction types, Let $p(inst|fid, l, tok)$ be the predicted probability distribution over all different instruction types by the parsing program, which is computed in the way described in Section 3.2. Based on current state of the LL machine, the LL machine provides a set of valid instruction types. Then for each instruction type, if it is in the set of valid instruction types, then its probability for sampling is $p(inst|fid, l, tok)$, otherwise its probability is set to be 0. Unless otherwise specified, at each step, the model chooses the instruction type predicted with the highest probability. The ways of predicting arguments for CALL and REDUCE instructions are similar.

We now give an analysis of $f(K)$, which is the total number of possible combinations of $c_1, ..., c_m$ given $m \leq K$. We consider $g(i)$ as the total number of $c_1, ..., c_i$ combinations for a fixed $i$, then we have

$$f(K) = \sum_{i=1}^{K} g(i) = \sum_{i=1}^{K} \frac{K!}{(K-i)!} = K!\left(\sum_{i=0}^{K-1} \frac{1}{i!}\right)$$

We now estimate it. In fact, we have that

$$exp(1) - \sum_{i=0}^{K-1} \frac{1}{i!} = \sum_{i=0}^{+\infty} \frac{1}{i!} - \sum_{i=0}^{K-1} \frac{1}{i!} = \sum_{i=K}^{+\infty} \frac{1}{i!} < \frac{1}{K!} + \frac{1}{K!}\frac{2}{K+1}$$

Also, we have

$$exp(1) - \sum_{i=0}^{K-1} \frac{1}{i!} > \frac{1}{K!}$$

Therefore, we have

$$0 \leq f(K) - K!exp(1) + 1 < \frac{2}{K+1} < 1$$

where $K \geq 2$. Therefore, we conclude $f(K) = \lfloor K!exp(1) - 1 \rfloor$.

## D   TRAINING DETAILS

Below we present full details about how to train the model. Following Section 4, we first illustrate the training approach when weak supervision is provided, and then explain how to train the model with input-output pairs only.

### D.1   WEAKLY SUPERVISED LEARNING

We assume that the set of all parameters is $\Theta$. We apply Adam optimizer to update

$$\Theta^{(i+1)} \leftarrow \Theta^{(i)} - \eta\Delta\Theta^{(i)}$$

where $\eta$ is the learning rate, and $\Delta\Theta^{(i)}$ is the gradient that consists of three components:

$$\Delta\Theta^{(i)} = \gamma_1 \cdot \Delta\Theta_1 + \gamma_2 \cdot \Delta\Theta_2 + \gamma_3 \cdot \Delta\Theta_3$$

In the following, we describe the three components $\Delta\Theta_1$, $\Delta\Theta_2$, and $\Delta\Theta_3$ respectively.

### D.1.1   REDUCE ARGUMENT $(c_1, ..., c_m)$

First, we present the details of $diff(T, T')$ in Algorithm 2. The first component of the gradient is computed as the following:

$$\Delta\Theta_1 = \sum_t \frac{\partial \log p(c_1^{(t)}, ..., c_m^{(t)}|n^{(t)})}{\partial \Theta} \cdot r_{\text{reduce}}(\hat{T}_r^{(t)})$$

where $t$ iterates over all REDUCE operations, $c_1^{(t)}, ..., c_m^{(t)}$ and $n^{(t)}$ indicate the predicted arguments in the $t$-th operation, and $\hat{T}_r^{(t)}$ indicates the predicted tree in the $t$-th operation.

**Algorithm 2** The algorithm to compute the difference between $T$ and $T'$. In the algorithm, we use $T = N(T_1, ..., T_j)$ to indicate that $T$'s root is non-terminal $N$, which has $j$ children $T_1, ..., T_j$.

> **function** $diff(T, T')$
> $\quad T = N(T_1, ..., T_j)$
> $\quad T' = N'(T'_1, ..., T'_{j'})$
> $\quad$**if** $N = N'$ **then**
> $\quad\quad sum \leftarrow 0$
> $\quad$**else**
> $\quad\quad sum \leftarrow 1$
> $\quad$**end if**
> $\quad$**if** $j < j'$ **then**
> $\quad\quad$**for** $i \leftarrow 1$ to $j$ **do**
> $\quad\quad\quad sum \leftarrow sum + diff(T_i, T'_i)$
> $\quad\quad$**end for**
> $\quad\quad$**for** $i \leftarrow j$ to $j'$ **do**
> $\quad\quad\quad sum \leftarrow sum + |T'_i|$
> $\quad\quad$**end for**
> $\quad$**else**
> $\quad\quad$**for** $i \leftarrow 1$ to $j'$ **do**
> $\quad\quad\quad sum \leftarrow sum + diff(T_i, T'_i)$
> $\quad\quad$**end for**
> $\quad\quad$**for** $i \leftarrow j'$ to $j$ **do**
> $\quad\quad\quad sum \leftarrow sum + |T_i|$
> $\quad\quad$**end for**
> $\quad$**end if**
> $\quad$**return** $sum$
> **end function**

### D.1.2 REDUCE ARGUMENT $n$

For learning to predict the REDUCE argument $n$, we can use reinforcement learning technique similar to the method above. In the following, we present another training method using supervised learning. We observe that such a training method is more time-efficient in our experiments.

We first match each REDUCE operation to a tentative ground truth. Given the predicted tree $\hat{T}$ and the ground truth $T_g$, we match each node in $\hat{T}$ to a node in $T_g$ in the following way. Assuming that $\hat{T} = \hat{N}(\hat{T}_1, ..., \hat{T}_k)$ and $T_g = N_g(T_{g1}, ..., T_{gk'})$, $\hat{N}$ is matched to $N_g$ first, then $\hat{T}_i$ is matched to $T_{gi}$ recursively for $i = 1, ..., \min(k, k')$. If $k > k'$, then $\hat{T}_i$ for $i \in \{k', ..., k\}$ is matched to any ground truth.

Afterwards, the second component is computed as follows:

$$\Delta\Theta_2 = \sum_t \frac{\partial \log p(n_g^{(t)}|fid^{(t)}, l^{(t)}, tok^{(t)})}{\partial\Theta}$$

where $t$ iterates over all REDUCE operations such that the generated non-terminal has a matched tentative ground truth $n_g^{(t)}$, and $\log p(n_g^{(t)}|fid^{(t)}, l^{(t)}, tok^{(t)})$ is the cross-entropy loss between $p(n^{(t)}|fid^{(t)}, l^{(t)}, tok^{(t)})$ and the one-hot vector of $n_g^{(t)}$.

### D.1.3 CALL ARGUMENT $fid$

We first give an example to illustrate our design of reward function $r_f$ in Figure 6.

The third component is computed as follows:

$$\Delta\Theta_3 = \sum_t \frac{\partial \log p(fid'^{(t)}|fid^{(t)}, l^{(t)}, tok^{(t)})}{\partial\Theta} \cdot r_f(fid'^{(t)})$$

where $t$ iterates over all CALL operations.

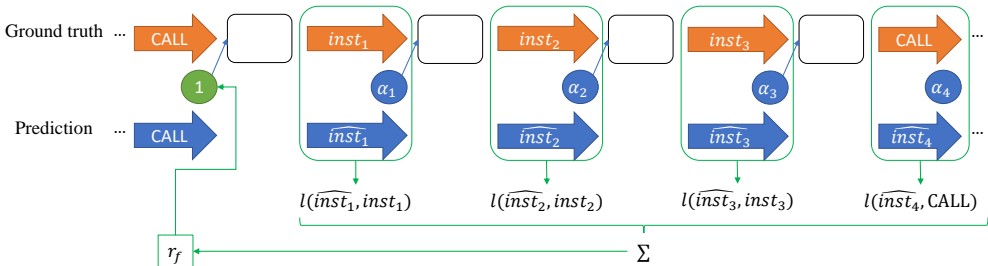

Figure 6: The illustration of the reward function $r_f$. The instructions colored orange indicate the ground truth, where none of $inst_1$, $inst_2$, and $inst_3$ is a CALL instruction. The reward function $r_f$ for the prediction of the first CALL's argument 1 (in the green circle) is the summation of four losses, where the loss function is the cross entropy loss.

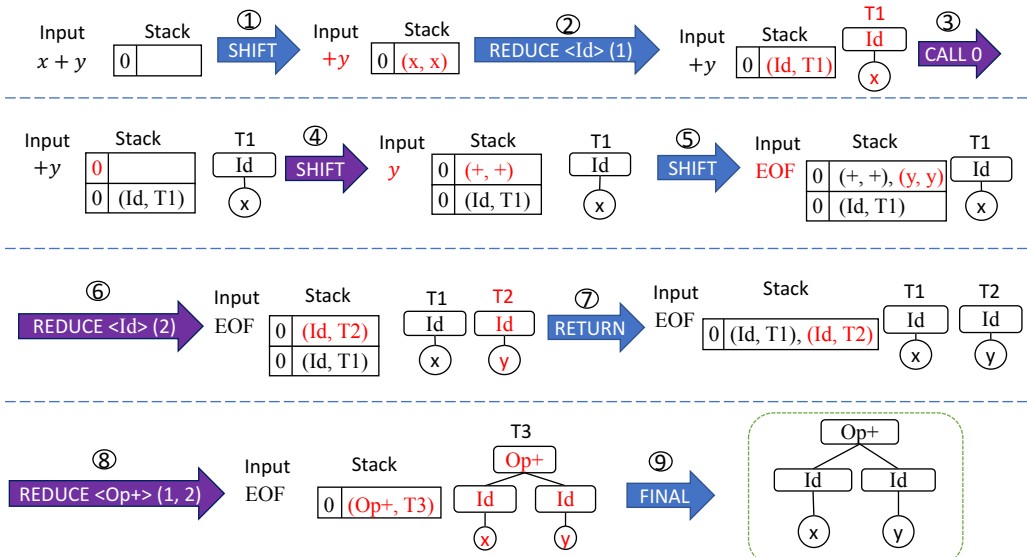

Figure 7: A wrong execution trace that can correctly parse x+y. The wrong operations (i.e., steps 3, 4, 6, and 8) are colored purple.

## D.2 TRAINING WITH INPUT-OUTPUT PAIRS ONLY

In this section, we further describe the algorithm for training with input-output pairs only, especially for how to search for the candidate trace set. As explained in Section 4.1, the algorithm needs to find the set of valid candidate traces for each input-output example. Notice that for one input-output example, the possible valid execution traces are not unique. Figure 7 provides one alternative execution trace that successfully parses x+y into its parse tree. Only when combining multiple examples, the model trained with this trace cannot fit all examples at the same time.

**Searching for the candidate trace set.** Here we further explain the two-nested-loop process to search for the candidate trace set following Section 4.1. First, in the outer loop, we randomly sample an instruction type trace based on the distribution described in Section 4.1. Then in the inner loop, we try to use the sampled trace in the external loop as the tentative ground truth, and then employ the weakly supervised learning approach to train the parameters predicting the arguments for $M_1$ iterations. If in any of these $M_1$ iterations, the correct output is produced, we add the sampled instruction trace to the candidate trace set. Otherwise, if the correct output is never produced during these $M_1$ iterations, we revert the model's parameters to predict the arguments back to those before these $M_1$ weak supervised learning iterations, and continue sampling another instruction trace. This

process is continued for $M_2$ iterations, i.e., a total of $M_2$ instruction traces are sampled. Meanwhile, to escape from a sub-optimal model, we re-initialize the model with the one learned from the previous lesson every $M_3$ iterations.

## E   HYPERPARAMETERS OF OUR PROPOSED METHOD

For the LL machines, $F = 10$. About the capacity of each stack frame $K$, $K = 3$ for WHILE language, and $K = 4$ for LAMBDA language. In the architecture of the neural parsing program, each LSTM has 1 layer, with its hidden state size $D = 50$, which is the same as the embedding size. As for the training, learning rate is $\eta = 0.01$ with no decay. No dropout is used. Gradient weights for the three components $\Delta\Theta_1$, $\Delta\Theta_2$ and $\Delta\Theta_3$ are $\gamma_1 = 10.0$, $\gamma_2 = 1.0$, and $\gamma_3 = 0.01$ respectively. Gradients with $L_2$ norm larger than 5.0 are scaled down to have the norm of 5.0. The model is trained using Adam optimizer. All weights are initialized uniformly randomly in $[-0.1, 0.1]$. The mini-batch size is 1. For candidate trace search, $\sigma = 0.1$, $M_1 = 20$, $M_2 = 10,000$, and $M_3 = 2,000$. Typically, for each input, the correct trace could be found after sampling within 1,000 traces.

## F   HYPERPARAMETERS OF BASELINE MODELS

For the baseline models in our evaluation, i.e., seq2seq Vinyals et al. (2015b), seq2tree Dong & Lapata (2016), and LSTM with unbounded memory Grefenstette et al. (2015), we implement them ourselves. We choose their hyperparameters based on their papers respectively, and further tune on our datasets to get better experimental results.

Specifically, in the seq2seq model Vinyals et al. (2015b), each of the encoder and the decoder is a 3-layer LSTM, and the hidden state size of each layer is 256, which is the same as the embedding size. We apply the attention mechanism described in Vinyals et al. (2015b). As for training, learning rate is 0.01. The dropout rate is 0.5. Gradients with $L_2$ norm larger than 5.0 are scaled down to have the norm of 5.0. The model is trained using Adam optimizer. All weights are initialized uniformly randomly in $[-0.1, 0.1]$. The mini-batch size is 256.

In the seq2tree model Dong & Lapata (2016), each of the encoder and the decoder is a 1-layer LSTM, and its hidden state size is 256, which is the same as the embedding size. We apply the attention mechanism described in Dong & Lapata (2016).As for training, learning rate is 0.005. The dropout rate is 0.5. Gradients with $L_2$ norm larger than 5.0 are scaled down to have the norm of 5.0. The model is trained using RMSProp optimizer. All weights are initialized uniformly randomly in $[-0.1, 0.1]$. The mini-batch size is 20.

In Stack-LSTM, Queue-LSTM and DeQue-LSTM models described in Grefenstette et al. (2015), each of the encoder and the decoder is a 1-layer LSTM, and its hidden state size is 256, which is the same as the embedding size. As for training, learning rate is 0.001. We do not use dropout for these models. Gradients with $L_2$ norm larger than 1.0 are scaled down to have the norm of 1.0. The model is trained using RMSProp optimizer. All weights are initialized uniformly randomly in $[-0.1, 0.1]$. The mini-batch size is 10.

## G   WHILE LANGUAGE

Below is the grammar specification of the WHILE language.

```
<Identifier>  ::=  x | y
   <Literal>  ::=  0 | 1
       <Op*>  ::=  <Identifier> × <Identifier>
               |   <Identifier> × <Literal>
               |   <Literal> × <Identifier>
               |   <Literal> × <Literal>
               |   <Op*> × <Identifier>
               |   <Op*> × <Literal>
       <Op+>  ::=  <Identifier> + <Identifier>
               |   <Identifier> + <Literal>
               |   <Identifier> + <Op*>
               |   <Literal> + <Identifier>
               |   <Literal> + <Literal>
               |   <Literal> + <Op*>
               |   <Op+> + <Identifier>
               |   <Op+> + <Literal>
               |   <Op+> + <Op*>
               |   <Op*> + <Identifier>
               |   <Op*> + <Literal>
               |   <Op*> + <Op*>
```

```
<Eq>  ::=  <Identifier> == <Identifier>
       |   <Identifier> == <Literal>
       |   <Identifier> == <Op+>
       |   <Identifier> == <Op*>
       |   <Literal> == <Identifier>
       |   <Literal> == <Literal>
       |   <Literal> == <Op+>
       |   <Literal> == <Op*>
       |   <Op+> == <Identifier>
       |   <Op+> == <Literal>
       |   <Op+> == <Op+>
       |   <Op+> == <Op*>
       |   <Op*> == <Identifier>
       |   <Op*> == <Literal>
       |   <Op*> == <Op+>
       |   <Op> == <Op*>
```

```
<Assign>  ::=  <Identifier> = <Identifier>
           |   <Identifier> = <Literal>
           |   <Identifier> = <Op+>
           |   <Identifier> = <Op*>
    <If>  ::=  <Assign> if <Identifier>
           |   <Assign> if <Literal>
           |   <Assign> if <Op+>
           |   <Assign> if <Op*>
           |   <Assign> if <Eq>
           |   <If> if <Identifier>
           |   <If> if <Literal>
           |   <If> if <Op+>
           |   <If> if <Op*>
           |   <If> if <Eq>
```

$$
\begin{array}{rcl}
\texttt{<Seq>} & ::= & \texttt{<Assign>;<Assign>} \\
& | & \texttt{<Assign>;<If>} \\
& | & \texttt{<Assign>;<While>} \\
& | & \texttt{<If>;<Assign>} \\
& | & \texttt{<If>;<If>} \\
& | & \texttt{<If>;<While>} \\
& | & \texttt{<While>;<Assign>} \\
& | & \texttt{<While>;<If>} \\
& | & \texttt{<While>;<While>} \\
& | & \texttt{<Seq>;<Assign>} \\
& | & \texttt{<Seq>;<If>} \\
& | & \texttt{<Seq>;<While>} \\
\texttt{<Block>} & ::= & \texttt{\{<Assign>\}} \\
& | & \texttt{\{<If>\}} \\
& | & \texttt{\{<While>\}} \\
& | & \texttt{\{<Seq>\}} \\
\texttt{<While>} & ::= & \textbf{while } \texttt{<Identifier> <Block>} \\
& | & \textbf{while } \texttt{<Literal> <Block>} \\
& | & \textbf{while } \texttt{<Op+> <Block>} \\
& | & \textbf{while } \texttt{<Op*> <Block>} \\
& | & \textbf{while } \texttt{<Eq> <Block>} \\
\end{array}
$$

## H   LAMBDA LANGUAGE

Below is the grammar specification of the LAMBDA language.

$$
\begin{array}{rcl}
\texttt{} & ::= & a \mid b \mid ... \mid z \\
\texttt{<App>} & ::= & \texttt{ } \\
& | & \texttt{<App> } \\
\texttt{<Bind>} & ::= & \textbf{lam } a \mid ... \mid \textbf{lam } z \\
\texttt{<Lam>} & ::= & \texttt{<Bind>. } \\
& | & \texttt{<Bind>. <App>} \\
& | & \texttt{<Bind>. <Lam>} \\
& | & \texttt{<Bind>. <Let>} \\
\texttt{<LetExpr>} & ::= & \texttt{ = } \\
& | & \texttt{ = <App>} \\
& | & \texttt{ = <Lam>} \\
& | & \texttt{ = <Let>} \\
\texttt{<Let>} & ::= & \textbf{let } \texttt{<LetExpr> } \textbf{in } \texttt{} \\
& | & \textbf{let } \texttt{<LetExpr> } \textbf{in } \texttt{<App>} \\
& | & \textbf{let } \texttt{<LetExpr> } \textbf{in } \texttt{<Lam>} \\
& | & \textbf{let } \texttt{<LetExpr> } \textbf{in } \texttt{<Let>} \\
\end{array}
$$

## I   PYTHON IMPLEMENTATION OF WHILE LANGUAGE PARSER

```python
def nextInstruction(self):
    fid, top = self.fid[-1], self.stack[-1]
    next = self.input[self.cur] if self.cur < len(self.input) else None
    if len(top) == 0:
        return self.shift, None
    elif len(top) == 1:
        if top[0][1] == 'while':
            return self.call, 0
        elif top[0][1] == '{':
            return self.call, 0
        elif top[0][1] == 'x' or top[0][1] == 'y':
            return self.reduce, (IDENT, [0])
        elif top[0][1] == '0' or top[0][1] == '1':
```

```
14                return self.reduce, (LIT, [0])
15            elif next == ';':
16                if fid < 1:
17                    return self.shift, None
18                else:
19                    return self.ret, None
20            elif next == 'if':
21                if fid < 2:
22                    return self.shift, None
23                else:
24                    return self.ret, None
25            elif next == '=':
26                if fid < 3:
27                    return self.shift, None
28                else:
29                    return self.ret, None
30            elif next == '==':
31                if fid < 4:
32                    return self.shift, None
33                else:
34                    return self.ret, None
35            elif next == '+':
36                if fid < 5:
37                    return self.shift, None
38                else:
39                    return self.ret, None
40            elif next == '*':
41                if fid < 6:
42                    return self.shift, None
43                else:
44                    return self.ret, None
45            elif next == None:
46                if len(self.stack) == 1:
47                    return self.final, None
48                else:
49                    return self.ret, None
50            else:
51                return self.ret, None
52        elif len(top) == 2:
53            if top[0][1] == '{' and next == '}':
54                return self.shift, None
55            else:
56                next_fid = 0
57                if top[0][1] == 'while':
58                    next_fid = 6
59                elif top[1][1] == ';':
60                    next_fid = 1
61                elif top[1][1] == 'if':
62                    next_fid = 2
63                elif top[1][1] == '=':
64                    next_fid = 3
65                elif top[1][1] == '==':
66                    next_fid = 4
67                elif top[1][1] == '+':
68                    next_fid = 5
69                elif top[1][1] == '*':
70                    next_fid = 6
71                return self.call, next_fid
72        else: # len(top) == 3
73            if top[1][1] == '=':
74                return self.reduce, (ASSIGN, [0, 2])
75            elif top[1][1] == 'if':
76                return self.reduce, (IF, [2, 0])
77            elif top[1][1] == ';':
78                return self.reduce, (SEQ, [0, 2])
```

```
79          elif top[0][1] == 'while':
80              return self.reduce, (WHILE, [1, 2])
81          elif top[0][1] == '{' and top[2][1] == '}':
82              return self.reduce, (BLOCK, [1])
83          else:
84              if top[1][1] == '+':
85                  return self.reduce, (OP_P, [0, 2])
86              elif top[1][1] == '*':
87                  return self.reduce, (OP_M, [0, 2])
88              elif top[1][1] == '==':
89                  return self.reduce, (EQ, [0, 2])
```

## J  PYTHON IMPLEMENTATION OF LAMBDA LANGUAGE PARSER

```
1  def nextInstruction(self):
2      fid, top = self.fid[-1], self.stack[-1]
3      next = self.input[self.cur] if self.cur < len(self.input) else None
4      if len(top) == 0:
5          return self.shift, None
6      elif len(top) == 1:
7          if top[0][1] == 'let':
8              return self.call, 0
9          elif top[0][1] == 'lam':
10             return self.shift, None
11         elif top[0][0] < 0 and top[0][1] in self.alpha:
12             return self.reduce, (VAR, [0])
13         else:
14             if next in self.alpha:
15                 if fid == 0:
16                     return self.call, 1
17                 else:
18                     return self.ret, None
19             elif next == '=' or next == '.':
20                 return self.shift, None
21             else:
22                 if len(self.stack) > 1:
23                     return self.ret, None
24                 else:
25                     return self.final, None
26     elif len(top) == 2:
27         if top[0][1] == 'let':
28             return self.shift, None
29         elif top[0][1] == 'lam':
30             return self.reduce, (BIND, [1])
31         elif top[1][1] == '=':
32             return self.call, 0
33         elif top[0][1] == BIND:
34             return self.call, 0
35         else:
36             return self.reduce, (APP, [0, 1])
37     elif len(top) == 3:
38         if top[0][1] == 'let' or top[0][1] == 'lam':
39             return self.call, 0
40         else:
41             nt = LETEXPR
42             if top[0][1] == BIND:
43                 nt = LAMBDA
44             return self.reduce, (nt, [0, 2])
45     else: # len(top) == 4:
46         return self.reduce, (LET, [1, 3])
```

## K   MORE ANALYSIS OF THE ROBUSTFILL DSL

**Efficiently enumerate the RobustFill space.**   RobustFill DSL allows programs to emit a concatenation of up to 10 constructors, each of which either outputs a constraint character, or the result of extracting a substring from the input and performing a string transformation such as replace, trim, etc. For each constructor, there can be approximately 30 million unique expressions, and thus the total number of programs with up to 10 constructors can be huge.

To efficiently enumerate the best expression, we rely on heuristics to cut most programs that do not lead to the best accuracy. In fact, if we enumerate the expression for each constructor from the first to the last, we need to require that the partial program generated should match the "most number of input-output examples". To this end, we manually construct a good program, and assume its accuracy is $p$. For each partial program, if it cannot yield a better accuracy by $p$, then the program will be dropped. That is, there are at least $(1 - p)N$ inputs that the program will result in an output that does not match the prefix of the ground truth.

Further, there will be empty constructors which will output an empty string for any input. They will cause the enumeration inefficient, and we also eliminate all empty constructors from being enumerated.

In doing so, we can estimate an upper bound on the accuracy that can be achieved by the RobustFill approach.

**Why RobustFill DSL cannot handle the parsing problem?**   In this section, we prove that the RobustFill DSL is not expressive for the parsing problem. In fact, the top-level constructor of the DSL allows an arbitrary number of tokens concatenated together. Since the RobustFill approach has the limitation to synthesize only programs with up to 10 tokens, we put this constraint to the DSL. Although this constraint looks artificial, this is not the fundamental reason for why the DSL is not expressive enough.

Each of these 10 tokens can be either a constant char, or a transformation of the substring of the input sequence. Therefore, the output of any program of the DSL can have up to 10 tokens. However, in the training set, each parse tree has more than 20 tokens, and thus no program can generate the parse tree. Note that the proof works for any programs with finite length. Given a program of length $n$, it cannot generate an output with more than $n$ tokens, but the test set can contain arbitrarily long outputs. Therefore, any specific program cannot generalize to longer outputs.

## L   MISCELLANEOUS

We present the three-line Python implementation of Quicksort below:

```
def qsort(a):
    if len(a) <= 1: return a
    return qsort([x for x in a if x<a[0]]) + \
           [x for x in array if x==a[0]] + qsort([x for x in a if x>a[0]])
```

