# OpenReview forum: "Towards Synthesizing Complex Programs From Input-Output Examples"
_ICLR.cc/2018/Conference — Accept (Poster)_

### Official Review · AnonReviewer3 · 2017-11-26
**Intriguing two phase RL approach for learning neural controllers for discrete programs**

**Rating:** 7
**Confidence:** 4

**Review:**

This paper presents a reinforcement learning based approach to learn context-free
parsers from pairs of input programs and their corresponding parse trees. The main
idea of the approach is to learn a neural controller that operates over a discrete
space of programmatic actions such that the controller is able to produce the
desired parse trees for the input programs. The neural controller is trained using
a two-phase reinforcement learning approach where the first phase is used to find
a set of candidate traces for each input-output example and the second phase is
used to find a satisfiable specification comprising of 1 unique trace per example
such that there exists a program that is consistent with all the traces. The
approach is evaluated on two datasets comprising of learning parsers for an
imperative WHILE language and a functional LAMBDA language. The results show that
the proposed approach is able to achieve 100% generalization on test sets with
programs upto 100x longer than the training programs, while baseline approaches
such as seq2seq and stack LSTM do not generalize at all.

The idea to decompose the synthesis task into two sub-tasks of first learning
a set of individual traces for each example, and then learning a program consistent
with a satisfiable subset of traces is quite interesting and novel. The use of
reinforcement learning in the two phases of finding candidate trace sets with
different reward functions for different operators and searching for a satisfiable
subset of traces is also interesting. Finally, the results leading to perfect
generalization on parsing 100x longer input programs is also quite impressive.

While the presented results are impressive, a lot of design decisions such as
designing specific operators (Call, Reduce,..) and their specific semantics seem
to be quite domain-specific for the parsing task. The comparison with general
approaches such as seq2seq and stack LSTM might not be that fair as they are
not restricted to only those operators and this possibly also explains the low
generalization accuracies. Can the authors comment on the generality of the
presented approach to some other program synthesis tasks?

For comparison with the baseline networks such as seq2seq and stack-LSTM, what
happens if the number of training examples is 1M (say programs upto size 100)?
10k might be too small a number of training examples and these networks can
easily overfit such a small dataset.

The paper mentions that developing a parser can take upto 2x/3x more time than
developing the training set. How large were the 150 examples that were used for
training the models and were they hand-designed or automatically generated by a
parsing algorithm? Hand generating parse trees for complex expressions seems to
be more tedious and error-prone that writing a modular parser.

The reason there are only 3 to 5 candidate traces per example is because the training
examples are small? For longer programs, I can imagine there can be thousands of bad
traces as it only needs one small mistake to propagate to full traces. Related to this
question, what happens to the proposed approach if it is trained with 1000 length programs?

What is the intuition behind keeping M1, M2 and M3 constants? Shouldn’t they be adaptive
values with respect to the number of candidate traces found so far?

For phase-1 of learning candidate traces, what happens if the algorithm was only using the
outside loop (M2) and performing REINFORCE without the inside loop?

The current paper presentation is a bit too dense to clearly understand the LL machine
model and the two-phase algorithm. A lot of important details are currently in the
appendix section with several forward references. I would suggest moving Figure 3
from appendix to the main paper, and also add a concrete example in section 4 to
better explain the two-phase strategy.

---

> ### Author Response · Authors · 2017-12-17
> **Response and revision plan**
>
> Thank you for your review!
>
> About the generality of our approach, we would like to mention that CALL, RETURN and FINAL instructions are general and used in the design of non-differentiable machines for a wide range of different programs, e.g., in Neural Programmer-Interpreter (NPI). The unique instructions of the LL machine are SHIFT and REDUCE, which are two fundamental operations to build a parser, and they are also used in Shift-Reduce machines proposed in NLP field. Although these two instructions are for parser per se, we would like to emphasize that the LL machine is generic enough to handle a wide spectrum of grammars. We will revise our paper to make this point clearer.
>
> For baseline models, we will train them on datasets with 1M samples including longer inputs. We may not be able to finish training on inputs of length 100 due to our hardware limitation, but will at least train on inputs of length 20, and we will update the results once we finish our experiments. However, we would like to point out that for some baseline models, it is already hard to fit to current training set with 10K samples. For example, the training accuracies of Stack-LSTM and DeQue-LSTM on LAMBDA dataset are below 3%, and the training accuracies of seq2seq on the two standard training sets are below 95%. We will open-source our code for replication after the double-blind review period. The code can also replicate the experiments in the original papers of the baseline models.
>
> For the training curriculum, the average lengths of training samples are 5.6 (Lambda) and 9.3 (WHILE), which are not long. Training samples in the curriculum are manually generated, which are used to test our manually written parser. The reason why generating a small set of training samples is faster than writing a parser is twofold. First, debugging the parser takes a long time, and it could take longer without the help of the training curriculum. Second, there are only a few samples in the curriculum, and these samples are short, thus it does not take long to generate them.
>
> Meanwhile, as we mentioned in our paper, our model relies heavily on the training curriculum to find the correct traces. In order to fit to longer samples, we need to train the model to fit to all samples in previous lessons with shorter samples first. At the beginning, the model can only find traces for the simplest input samples, e.g., x + y. Then the model gradually learns to fit to samples of length 5, 7, etc., in the training curriculum. If we randomly initialize the model and then train it directly on samples of length 1000, then our model will completely fail to find any trace that leads to the correct output parse tree. We will update our paper to explain more about the curriculum as well.
>
> The choice of hyper-parameters M1, M2 and M3 is based on our empirical results. Their values are not adaptively tuned to make sure that the training algorithm can search for candidate traces for enough time. For example, we can not simply stop the algorithm after finding 3 candidate traces, because we are not sure whether it can still find the 4th trace or more.
>
> For the training algorithm, without the inner loop in phase 1, the model will trap into a local minimum without finding the whole set of traces. Most likely, the traces found in this case are wrong ones.
>
> Thank you for your advice on writing! We defer these details to the Appendix to shorten the main body of our paper. Following your suggestions, we will add a section to include some running examples, and provide more descriptions of our training algorithm. Also, we will move some important details in the Appendix to the main body of the paper.

---

> > ### Author Response · Authors · 2017-12-30
> > **We have added results of baseline models trained on 1M samples**
> >
> > We have added a set of experiments to train baseline models on the dataset with 1M samples of length 50, and included the results in our paper. We observe that for seq2seq and seq2tree models, training with 1M samples mitigates the overfitting issue; however, for Stack LSTM, Queue LSTM and DeQue LSTM, both training and test accuracies drop to 0.

---

### Official Review · AnonReviewer2 · 2017-11-27
**good paper – some clarifications would help**

**Rating:** 8
**Confidence:** 3

**Review:**

Summary:
I thank the authors for their update and clarifications.  They have addressed my concerns, and I will keep my score as it is.

-----------------------------------------------------------------------
The authors present a system that parses DSL expressions into syntax trees when trained using input-output examples. Their approach is based around LSTMs predicting a sequence of program-like instructions/arguments, and they argue that their work is an illustration of how we should approach synthesis of complex algorithms using neural techniques.

Overall I liked this paper:
The authors provide a frank view on the current state of neural program synthesis, which I am inclined to agree with: (1) existing neural program synthesis has only ever worked on ‘trivial’ problems, and (2) training program synthesizers is hard, but providing execution traces in the training data is not a practical solution. I am somewhat convinced that the task considered in this paper is not trivial (so the authors do not obviously fall into trap (1) ), and I am convinced that the authors’ two-phase reinforcement learning solution to (2) is an interesting approach.
My score reflects the fact that this paper seems like a solid piece of work: The task is difficult, the solution interesting and the results are favourable.

However, I would like the authors to clarify the following points:
1.	In the inner loop of Algorithm 1, it looks like Net_in is updated M1 times, and a candidate trace is only stored if arguments that generate an exact match with the ground truth tree are found. Since M1 = 20, I am surprised that an exact match can be generated with so few samples/updates. Similarly, I am surprised that the appendix mentions that only 1000 samples in the outer loop are required to find an exact match with the instruction trace. Could you clarify that I am reading this correctly and perhaps suggest intuition for why this method is so efficient. What is the size of the search space of programs that your LL machine can run? Should I conclude that the parsing task is actually not as difficult as it seems, or am I missing something?
2.	The example traces in the appendix (fig 3, 6) only require 9 instructions. I’m guessing that these short programs are just for illustration – could you provide an example execution trace for one of the programs in the larger test sets? I assume that these require many more instructions & justify your claims of difficulty.
3.	As a technique for solving the parsing problem, this method seems impressive. However, the authors present the technique as a general approach to synthesizing complex programs. I feel that the authors need to either justify this bold assertion with least one additional example task or tone down their claims. In particular, I would like to see performance on a standard benchmarking task e.g. the RobustFill tasks. I want to know whether (1) the method works across different tasks and (2) the baselines reproduce the expected performance on these benchmark tasks.
4.	Related to the point above, the method seems to perform almost too well on the task it was designed for – we miss out on a chance to discuss where the model fails to work.

The paper is reasonably clear. It took a couple of considered passes to get to my current understanding of Algorithm 1, and I found it essential to refer to the appendix to understand LL machines and the proposed method. In places, the paper is somewhat verbose, but since many ideas are presented, I did not feel too annoyed by the fact that it (significantly) overshoots the recommended 8 page limit.

---

> ### Author Response · Authors · 2017-12-17
> **Clarification of the complexity of our parsing problem and revision plan**
>
> Thank you for your review! We will upload a revision by next week. We will update our paper to explain more about our training method. In particular, we will include some running examples to further explain our training algorithm and demonstrate the complexity of our parsing problem. About your questions:
>
> 1. We will add more explanation about what will happen during training. To give you a teaser, consider an example input of 3 tokens, x+y, there are 72810 valid traces --- which seems not so many. But consider that we need to find one valid trace for each example, then there could be 72810^n combinations for a training set of n examples of length 3. In this sense, even n>=2 will render an exhaustive approach impractical. When the input length increases to 5 tokens, the number of valid traces increases to 50549580, and 33871565610 for 7. We can observe that the search space grows exponentially as input length increases. We hope this can give you an intuitive idea of the difficulty of the problem that we are working on. We will provide a more detailed explanation of the problem complexity in our revision.
>
> 2. The challenge does not come from the length of the execution trace --- it is mostly linear to the length of the input and output. The main difficulty comes from the volume of the search space, which can grow exponentially large as the length of the execution trace increases.
>
> 3. We believe our technique can be generalized to other tasks, but the evaluation on other tasks will not be easy. The main challenge is that we need to design a new domain-specific machine, a neural program architecture, so that we can test our RL-based strategy. This could result in a workload of an entirely new research. Notice that it is also not trivial to adapt RobustFill to other tasks, as we mentioned in our paper. Meanwhile, since Microsoft will not release the FlashFill test data, it can be hard to make a fair comparison on their task. Thus, we will choose to tone down our claim and make our contribution more specific to the parsing problem.
>
> 4. We will add a section to include some running examples and failure modes in our revision, either in the main body or in the appendix depending on what reviewers prefer.

---

> > ### Comment · AnonReviewer2 · 2018-01-04
> > **How much ambiguity?**
> >
> > I'm still not sure I understand how much ambiguity is covered by the second phase of the algorithm.  For example, in the case of your AM language, how many traces are there for the input "x+y" that yield the correct parse tree.  If I eliminate traces with degenerate call/return pairs, and ignore the function ids then I can only see two different traces which generate the correct parse tree (the one in the paper, plus one which adds an additional call return pair before shifting the plus).  Adding the function ids back in, with K=3, I think there is still only 12 possible correct traces.  Would it be possible to calculate the average number of correct traces for each parse tree in the training set for each of your 6 evaluation settings?  Would it be possible to calculate it for each of the training samples in your toy AM language (i.e. each of the samples at the bottom of page 10) for illustrative purposes?  It would be great if you could also calculate the number of correct instruction type traces as well.  This would really help to clarify the inherent difficultly of the underlying problem.

---

> > > ### Author Response · Authors · 2018-01-05
> > > **Further clarifications**
> > >
> > > We want to clarify that the main challenge for phase 2 is the large search space of the valid specifications due to the rule of product. In particular, in AM language, assuming we find 3 candidate traces leading to the correct output for each training example, then the search space of phase 2 is 3^24=282,429,536,481, which is very large. Our solution first breaks the entire training set into several smaller lessons, so that the search space is reduced (see the "The effectiveness of training curriculum" paragraph in Sec 5, p. 13); also, in phase II of the algorithm, we use a sampling-based strategy to further reduce the number of specifications examined to be within 30 (see the "Phase II: Searching for a satisfiable specification." paragraph in Sec 4.1, p. 9).
> > >
> > > Therefore, the small number of candidate traces for each single input-output example does not make the problem simple, since the large search space of our problem mainly comes from the rule of product, which is the major difficulty.
> > >
> > > Having said this, we would like to clarify some further questions. For the "x+y" example, there are 3 shortest correct instruction type traces: 2 are provided in Fig 2 and 7, and the third (wrong) one is as follows:
> > >
> > > SHIFT SHIFT REDUCE CALL SHIFT REDUCE RETURN REDUCE FINAL
> > >
> > > In our experiments, we find that each input-output example has at least 3-5 candidate instruction type traces. But unfortunately, we cannot provide the full set of instruction type traces for our WHILE and LAMBDA training curriculum. To do so, we need to exhaustively enumerate all possible instruction traces from a huge space that is intractable for even inputs of length 9. We do not know any practical approaches to estimate this number so far.
> > >
> > > We also want to clarify that our candidate traces involve the minimal number of call/returns necessary for the LL machine to produce the correct output. In fact, we can add arbitrary number of call/returns to make a trace still valid; however, doing so will make the trace longer, so they no longer have the minimal length.
> > >
> > > We further add the number of correct execution traces and correct instruction type traces of the shortest length for each example in the AM training set at the top of page 13. Hope these can help to clarify your concerns.

---

### Official Review · AnonReviewer1 · 2017-11-28
**Interesting work with strong results, but lacks empirical analysis**

**Rating:** 5
**Confidence:** 2

**Review:**

This paper proposes a method for learning parsers for context-free languages. They demonstrate that this achieves perfect accuracy on training and held-out examples of input/output pairs for two synthetic grammars. In comparison, existing approaches appear to achieve little to no generalization, especially when tested on longer examples than seen during training.

The approach is presented very thoroughly. Details about the grammars, the architecture, the learning algorithm, and the hyperparameters are clearly discussed, which is much appreciated. Despite the thoroughness of the task and model descriptions, the proposed method is not well motivated. The description of the relatively complex two-phase reinforcement learning algorithm is largely procedural, and it is not obvious how necessary the individual pieces of the algorithm are. This is particularly problematic because the only empirical result reported is that it achieves 100% accuracy. Quite a few natural questions left unanswered, limiting what readers can learn from this paper, e.g.
- How quickly does the model learn? Is there a smooth progression that leads to perfect generalization?
- Presumably the policy learned in Phase 1 is a decent model by itself, since it can reliably find candidate traces. How accurate is it? What are the drawbacks of using that instead of the model from the second phase? Are there systematic problems, such as overfitting, that necessitate a second phase?
- How robust is the method to hyperparameters and multiple initializations? Why choose F = 10 and K = 3? Presumably, there exists some hyperparameters where the model does not achieve 100% test accuracy, in which case, what are the failure modes?

Other misc. points:
- The paper mentions that "the training curriculum is very important to regularize the reinforcement learning process." Unless I am misunderstanding the experimental setup, this is not supported by the result, correct? The proposed method achieves perfect accuracy in every condition.
- The reimplementations of the methods from Grefenstette et al. 2015 have surprisingly low training accuracy (in some cases 0% for Stack LSTM and 2.23% for DeQueue LSTM). Have you evaluated these reimplementations on their reported tasks to tease apart differences due to varying tasks and differences due to varying implementations?

---

> ### Author Response · Authors · 2017-12-17
> **Clarification of some misunderstandings and revision plan**
>
> Thank you for your review! We are working on a revision, and we will upload the new version no later than next week.
>
> We will update our paper to explain more about our training method, including the curriculum learning and two-phase training algorithm. In particular, we will add a section to include some running examples, and describe what would happen if not following our proposed strategy, e.g., removing phase 2 in our algorithm. In general, using an alternative method, the model could overfit to a subset of the training examples, and thus fails to generalize.
>
> For the choice of hyper-parameters, F (the number of different function IDs) and K (the maximal number of elements in each stack frame’s list) are parameters of the LL machine, not the training algorithm. These parameters ensure that the LL machine is expressive enough to serve as a parser. If the values of them are too small, then there exists no program that can run the machine to simulate the parser for a complex grammar.
>
> For the curriculum learning, we want to point out that we only report our approach employing the curriculum training. If curriculum training is not employed, then our model cannot even fit to the training data, and thus will fail completely on the test data. This point has been explained in Section 4.1 and Section 5. We will make it more explicit in our next revision.
>
> Our re-implementation of the methods from Grefenstette et al. 2015 can replicate their results on their tasks. We will open-source our code for replication of both Grefenstette et al. 2015 and our experiments after the double-blind review period.

---

### Public Comment · (anonymous) · 2017-12-11
**Only parsing ?**

The paper presents an impressive result on the task of learning a parser, but those achievements rest upon the design of a task-specific "machine", specifically devised for parsing. This blatantly misses the point of program induction (which the title suggests is the ultimate goal, i.e. learning "complex programs from input-output examples"). Maybe am I mistaken and learning a parser is all the authors are interested in ? In that case the title is misleading at best.

Otherwise, is the designed "machine" Turing-complete, such that there is a remote possibility of it learning non-parsing tasks ? Such a prospective extension of this work would anyway warrant additional experiments and supporting evidence. The operation of the "machine" for performing other tasks than parsing would likely require different inductive biases.

---

> ### Author Response · Authors · 2017-12-17
> **Learning programs on domain-specific machines is as important as learning programs on generic machines**
>
> Thank you for your comment! We would like to clarify this common confusion.
>
> We want to emphasize that we highly agree that the learning neural programs on a generic machine, such as Turing Machine, is an important area. However, in many cases, such a problem can be hard or even implausible. For example, consider the problem of sorting, when only an unsorted list is given as the input, and the corresponding sorted list is given as the output, how is it possible to know whether the target program to be learned is a merge sort or a quick sort or any other sorting algorithm? Such an ambiguity cannot be easily resolved without providing more specification to restrict the search space.
>
> In particular, another important field is to restrict the search space to be within a concrete domain by customizing the domain-specific machine. In doing so, the program to be learned can be restricted to the domain of interest as well. We have observed many important papers adopting this style. For example, two best papers from ICLR 2016 and ICLR 2017 are studying Neural Programmer-Interpreter (NPI). In NPI, each different task uses a different domain-specific machine; one of the latest program synthesis work, RobustFill (ICML 2017), also restricts its programs to be in the domain defined by a domain-specific language composed by string operations. These are all successful applications of applying a domain-specific machine/language to restrict the program of interest to be learned. Our work is following the same paradigm, but jumps a big step forward to consider a much more complex program domain.
>
> To sum up, we agree that learning programs on a generic machine is important; but we also want to argue and highlight that learning programs on a domain specific machine may have more practical impact, and is definitely attracting more interests.

---

### Author Response · Authors · 2017-12-23
**A revision of the paper has been uploaded**

We have updated the paper with the following changes:

(1) We add a section (Section 5) including a running example to further describe the training curriculum, motivate the two-phase algorithm, and present the search space to show the difficulty of our parsing problem.

(2) We move a figure illustrating the parsing process with the LL machine from the Appendix to the main body, i.e., Figure 2 in the revised version.

(3) We have carefully revised our claim throughout the paper to mention that our strategy is only evaluated to be effective on the parser learning task. We also mention that we leave applying this strategy to more tasks as a future direction.

---

### Decision · Program_Chairs · 2018-01-29
**ICLR 2018 Conference Acceptance Decision**

**Decision:**

Accept (Poster)

**Comment:**

This paper proposes a method for training an neural network to operate stack-based mechanism in order to act as a CFG parser in order to, eventually, improve program synthesis and program induction systems. The reviewers agreed that the paper was compelling and well supported empirically, although one reviewer suggested that analysis of empirical results could stand some improvement. The reviewers were not able to achieve a clear consensus on the paper, but given that the most negative reviewer has also declared themselves the least confident in their assessment, I am happy to recommend acceptance on the basis of the median rather than mean score.